# Can LLMs Outshine Conventional Recommenders?
# A Comparative Evaluation

**Qijiong Liu[1], Jieming Zhu[2], Lu Fan[1], Kun Wang[3],**
**Hengchang Hu[4], Wei Guo[2], Yong Liu[2], Xiao-Ming Wu[1]✉**
[1]PolyU Hong Kong [2]Huawei Noah's Ark Lab [3]NTU Singapore [4]NUS Singapore
liu@qijiong.work, xiao-ming.wu@polyu.edu.hk

## Abstract

Integrating large language models (LLMs) into recommender systems has created new opportunities for improving recommendation quality. However, a comprehensive benchmark is needed to thoroughly evaluate and compare the recommendation capabilities of LLMs with traditional recommender systems. In this paper, we introduce RECBENCH, which systematically investigates various item representation forms (including unique identifier, text, semantic embedding, and semantic identifier) and evaluates two primary recommendation tasks, i.e., click-through rate prediction (CTR) and sequential recommendation (SeqRec). Our extensive experiments cover up to 17 large models and are conducted across five diverse datasets from fashion, news, video, books, and music domains. Our findings indicate that LLM-based recommenders outperform conventional recommenders, achieving up to a 5% AUC improvement in CTR and up to a 170% NDCG@10 improvement in SeqRec. However, these substantial performance gains come at the expense of significantly reduced inference efficiency, rendering LLMs impractical as real-time recommenders. We have released our code[1] and data[2] to enable other researchers to reproduce and build upon our experimental results.

## 1 Introduction

Recommender systems are essential for providing personalized information to internet users. The design of these systems typically involves balancing objectives such as fairness, diversity, and interpretability. In industrial applications, however, accuracy and efficiency are paramount. Accuracy underpins user experience, greatly influencing user satisfaction and engagement, while efficiency is crucial for timely recommendation delivery and system deployment.

In recent years, the integration of large language models (LLMs) into recommender systems (denoted as LLM+RS) has garnered significant attention from academia and industry. These integrations fall into two main paradigms [1, 2, 3, 4]: LLM-FOR-RS and LLM-AS-RS. LLM-FOR-RS enhances traditional deep learning-based recommender models (DLRMs) through advanced feature engineering or encoding techniques using LLMs [5, 6]. This paradigm acts as a plug-in module, easily integrating with existing recommender systems, offering high efficiency, and improving accuracy without substantial overhead, making it ideal for industrial applications. Conversely, LLM-AS-RS employs LLMs directly as recommenders to generate recommendations. This paradigm excels in specific contexts, such as cold-start scenarios [7], and tasks requiring natural language understanding and generation, like interpretable and interactive recommendations [8, 9, 10]. Despite its potential, the extremely low

---

[1]https://recbench.github.io
[2]https://www.kaggle.com/datasets/qijiong/recbench/

39th Conference on Neural Information Processing Systems (NeurIPS 2025) Track on Datasets and Benchmarks.

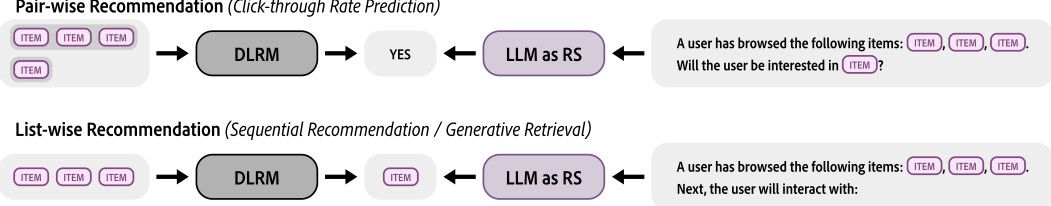

Figure 1: Illustration of DLRM and LLM recommender in two scenarios. Each ITEM represents a placeholder that can be filled with various item representations, including *unique identifier*, *text*, *semantic embedding* or *semantic identifier*.

inference efficiency of large models poses challenges for high-throughput recommendation tasks. Nevertheless, LLM-AS-RS is reshaping traditional recommendation pipeline designs.

Several benchmarks have been proposed for LLM-AS-RS, including LLMRec [11], PromptRec [12], and others [13, 14, 15]. However, as shown in Table 1, these benchmarks have limitations: (i) they often evaluate only a single recommendation scenario, (ii) their coverage of item representation forms for alignment within LLMs is narrow, typically limited to conventional *unique identifier* or *text* formats, and (iii) they assess a relatively small number of traditional models, large-scale models, and datasets, leading to an incomplete and fragmented performance landscape in this domain.

To address existing gaps, we propose the RECBENCH platform for a comprehensive evaluation of the LLM-AS-RS paradigm. **Firstly,** we explore various item representation and alignment methods between recommendation scenarios and LLMs, including *unique identifier*, *text*, and *semantic embedding*, and *semantic identifier*, to assess their impact on recommendation performance. **Secondly,** the benchmark covers two main recommendation tasks: *click-through rate (CTR) prediction and sequential recommendation (SeqRec)*, representing pair-wise and list-wise recommendation scenarios, respectively. **Thirdly,** our study evaluates up to **17** LLMs, encompassing general-purpose models like Llama [16]) and recommendation-specific models like RecGPT [17]. This extensive evaluation supports multidimensional comparisons across models of varying sizes (e.g., $OPT_{base}$ and $OPT_{large}$), from different institutions (e.g., Llama and Qwen), and different versions from the same institution (e.g., $Llama-1_{7B}$ and $Llama-3_{8B}$). **Fourthly,** experiments are conducted across **5** recommendation datasets from *diverse domains*–including fashion (HM [18]), news (MIND [19]), video (MicroLens [20]), books (Goodreads [21]), and music (Amazon CDs [22])–to ensure balanced comparisons without reliance on a single platform. **Fifthly,** we assess both *recommendation accuracy and efficiency*, providing a holistic comparison between conventional DLRMs and LLM-AS-RS. Our evaluation includes *zero-shot and fine-tuning* approaches: zero-shot examines LLMs' inherent recommendation and reasoning abilities, while fine-tuning evaluates their adaptability in new scenarios.

In summary, our RECBENCH benchmark provides a comprehensive evaluation of the LLM-AS-RS paradigm, revealing several key insights: **Firstly**, while LLM-based recommenders show significant performance improvements across various scenarios, their efficiency limitations hinder practical deployment. Future research should prioritize developing inference acceleration techniques for LLMs in recommendations. **Secondly**, conventional DLRMs enhanced with LLM support (LLM-FOR-RS paradigm, Group C in Figure 2) can achieve up to 95% of the performance of standalone LLM recommenders while operating much faster. Thus, enhancing the integration of LLM capabilities into conventional DLRMs is a promising research direction. We hope that our established, reusable, and standardized RECBENCH will lower the evaluation barrier and accelerate the development of new models within the recommendation community.

## 2   Preliminaries and Related Work

We first introduce various forms of item representation, which is the foundation of recommender systems. Then, we present a unified evaluation framework (Fig. 1). Next, we review representative work to highlight current advancements. Finally, we compare our RECBENCH with existing benchmarks.

**Item Representations.** Item representation is a critical component of recommender systems. Since the introduction of deep learning in this field, the most prevalent approach [23, 24, 25] has been to

Table 1: Comparison of RECBENCH with existing benchmarks within the LLM-AS-RS paradigm. "–" indicates that, despite its claims, LLMRec does not practically support list-wise recommendation.

| Benchmark
Year | | Zhang et al.
2021 | OpenP5
2024 | LLMRec
2023c | PromptRec
2024b | Jiang et al.
2024 | RSBench
2024b | RECBENCH
(ours) |
|---|---|---|---|---|---|---|---|---|
| **Scale** | #DLRM | 2 | 9 | 13 | 4 | 6 | 0 | 10 |
| | #LLM | 4 | 2 | 7 | 4 | 7 | 1 | 17 |
| | #Dataset | 1 | 10 | 1 | 3 | 4 | 3 | 5 |
| **Scheme** | Zero-shot | ✓ | × | ✓ | ✓ | × | ✓ | ✓ |
| | Fine-tune | ✓ | ✓ | ✓ | ✓ | ✓ | × | ✓ |
| **Item
Representation** | *unique identifier* | × | ✓ | ✓ | × | × | × | ✓ |
| | *text* | ✓ | × | ✓ | ✓ | ✓ | ✓ | ✓ |
| | *semantic embedding* | × | × | × | × | × | × | ✓ |
| | *semantic identifier* | × | × | × | × | × | × | ✓ |
| **Scenario** | Pair-wise | × | ✓ | ✓ | ✓ | ✓ | ✓ | ✓ |
| | List-wise | ✓ | ✓ | – | × | × | × | ✓ |
| **Metric** | Quality | ✓ | ✓ | ✓ | ✓ | ✓ | ✓ | ✓ |
| | Efficiency | × | × | × | × | × | × | ✓ |

use item *unique identifier*. These identifiers initially lack intrinsic meaning, and their corresponding vectors are randomly initialized and will be learned from user–item collaborative signals.

With advancements in computational power and the advent of the big data era, item content–such as product images and news headlines–has increasingly been utilized for item representation. Simple modules like convolutional neural networks [26] and attention networks [27] are designed to encode item *text* into unified item representations based on *text* for the recommendation models.

In recent years, the pretrained and open-source language models are widely integrated with the recommendation model and served as the end-to-end item encoder, fine-tuned with the recommendation tasks. The *semantic embedding* has been proven to be more effective than previous shallow networks with *text*, as the former introduce rich general semantics into the recommendation model [28, 29].

Additionally, a new form of item representation: *semantic identifier*, is introduced recently. With semantic embeddings obtained from LLMs, discrete encoding techniques like RQ-VAE [30] are used to map items into unique, shareable identifier combinations. Items with similar content are characterized by longer common subsequences. The use of *semantic identifier* not only efficiently compresses the item vocabulary but also maintains robust semantic relationships during training [31].

The emergence and advantages of the semantic identifier have reshaped sequential recommendation methods, also known as generative retrieval [32, 33, 34, 35]. They provide new input forms and alignment strategies between LLMs and recommender systems, paving the way for advancements in the LLM-AS-RS paradigm [36, 31].

**Recommendation Scenarios Evaluated.** As LLMs exhibit significant reasoning capabilities across various domains [37, 38, 39], the recommendation community is exploring their direct application to recommendation tasks [40, 41]. This LLM-AS-RS paradigm abandons conventional DLRMs, seeking to harness the robust semantic understanding and deep Transformer architectures of LLMs to capture item features and model user preferences and generate recommendation results. To assess how LLMs operate within this paradigm, we examine two common recommendation scenarios (Fig. 1):

*Pair-wise Recommendation*, also known as straightforward recommendation [42], corresponds to the traditional Click-Through Rate (CTR) prediction task [23, 24]. The input consists of a user-item pair, and the LLM is expected to output a recommendation score for this pair (e.g., the predicted likelihood that the user will click on the item).

*List-wise Recommendation* typically corresponds to sequential recommendation tasks [43, 44]. The input comprises a sequence of items with positive feedback from a user, and the LLM is expected to predict the next item that the user is likely to engage with. In contrast to DLRMs that use structured feature inputs, the LLM-AS-RS paradigm requires concatenating inputs in natural language and guiding the LLM to generate the final results.

**LLMs as Recommender Systems.** The progression of LLM-AS-RS can be divided into three stages:

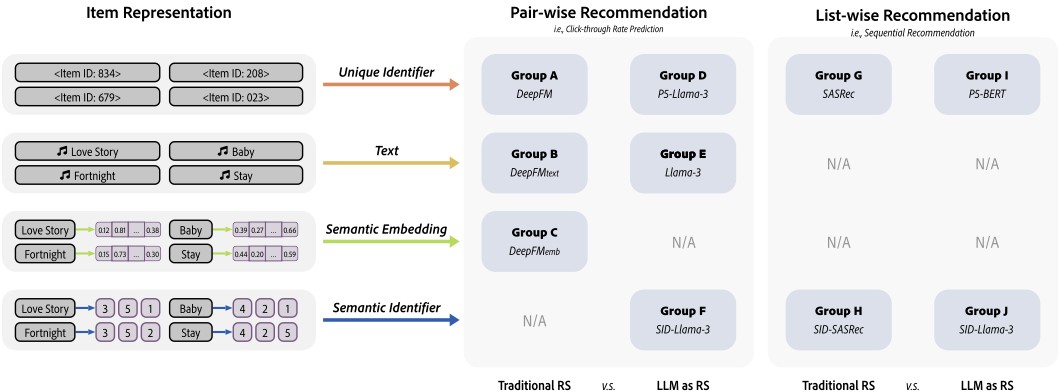

Figure 2: (Left) Various forms of item representations. (Right) Groups chosen for benchmarking and their representative methods. N/A: there is no or few related work, and it will not be evaluated.

*Stage One: Zero-shot Recommendations with LLMs.* Early studies explored whether general-purpose LLMs could perform recommendations without fine-tuning. While their performance lagged behind traditional DLRMs [13, 40, 41, 45], they outperformed random baselines, revealing limited yet meaningful recommendation capabilities. As LLMs could only process textual input, *text* served as the universal item representation across domains, bridging LLMs and recommender systems.

*Stage Two: Fine-Tuning LLMs for Recommendation.* This stage adapted LLMs to recommendation tasks via supervised fine-tuning. Approaches like Uni-CTR [46] and Recformer [47] aligned recommender systems with LLMs using semantic text in pair-wise learning. Others, such as P5 [48, 42] and VIP5 [49], introduced non-textual signals (e.g., item *unique identifier*) for multi-task learning on Amazon data. Models like LLaRA [50] and LLM4IDRec [51] further advanced LLMs' ability to model user behavior through sequential recommendation fine-tuning.

*Stage Three: Integration of Semantic Identifiers with LLMs.* Recently, researchers combined *semantic identifier* with LLMs to enhance recommendation performance [52]. For instance, LC-Rec [36] extended the multi-task learning paradigm of P5 but replaced item representations with semantic identifiers, achieving breakthrough results. STORE [31] introduced a parallel semantic tokenization approach that each token represents a unique perspective of item content.

**Comparison with Previous Benchmarks.** Table 1 provides a summary of existing benchmarks within the LLM-AS-RS paradigm. [13] was a pioneering effort in using language models for sequential recommendation, evaluating BERT's recommendation capabilities in both zero-shot and fine-tuning scenarios using the MovieLens dataset [53]. This work marks the inception of research into using LLMs directly as recommender systems.

Building on the P5 method [48], OpenP5 [42] evaluated multiple recommendation scenarios alongside conventional methods, albeit using only unique identifiers for item representation. PromptRec [12] focused on cold-start scenarios, comparing LLMs with conventional deep learning recommendation models (DLRMs) using solely semantic text for zero-shot recommendation, highlighting the advantages of LLMs in content understanding. The study by [14] employed multidimensional evaluation metrics but fine-tuned LLMs exclusively using semantic text. RSBench [15] primarily optimized for conversational recommendation scenarios but utilized only a single LLM and lacked comparisons with traditional recommendation models. LLMRec [11] emulated P5 by training LLMs through multitask learning and employed both unique identifiers and semantic text as item representations. However, LLMRec did not incorporate semantic identifiers into item representations and conducted experiments solely on the Amazon Beauty dataset [22], which limits its generalizability.

Our RECBENCH provides a comprehensive evaluation of the recommendation capabilities of **17** LLMs across **5** datasets. Utilizing **4** different forms of item representation and assessed in both *pair-wise and list-wise* recommendation scenarios, our benchmark uniquely evaluates the efficiency of recommendation models, aligning with the principles of Green AI [54] in the era of large models.

# 3 Proposed Benchmark: RECBENCH

This section details our benchmarking approaches for two main recommendation scenarios.

## 3.1 Pair-wise Recommendation

Pair-wise recommendation estimates the probability $\hat{y}_{u,t}$ that a user $u$ interacts with (e.g., clicks on) an item $t$. Models are typically trained with binary cross-entropy loss:

$$\mathcal{L} = -\sum_{(u,t)\in\mathcal{D}} \left[ y_{u,t} \log \hat{y}_{u,t} + (1 - y_{u,t}) \log(1 - \hat{y}_{u,t}) \right], \tag{1}$$

where $\mathcal{D}$ is the set of all user–item interactions. As illustrated in Figure 1, we use user behavior sequence as the user-side feature.

**Group A: Deep CTR models with *unique identifier*.** For these models, each item embedding $\mathbf{t}$ is randomly initialized. A user is represented by averaging the embeddings of items in their behavior sequence:

$$\mathbf{u} = \frac{1}{N_u} \sum_{i=1}^{N_u} \mathbf{t}_{u_i}, \tag{2}$$

where $N_u$ is the sequence length. The CTR model $\Phi$ predicts the click probability as:

$$\hat{y}_{u,t} = \Phi\left(\mathbf{u}, \mathbf{t}\right). \tag{3}$$

The models selected for benchmarking are DNN, PNN [55], DCN [23], DCNv2 [23], DeepFM [24], MaskNet [56], FinalMLP [57], AutoInt [58], and GDCN [59].

**Group B: Deep CTR models with *text*.** These models learn item representations from textual features:

$$\mathbf{t} = \frac{1}{N_t} \sum_{i=1}^{N_t} \mathbf{w}_{t_i}, \tag{4}$$

where $N_t$ is the text sequence length, and $\mathbf{w}_t$ denotes the item text sequence embeddings. The models selected for benchmarking include $\text{DNN}_{\text{text}}$, $\text{DCNv2}_{\text{text}}$, $\text{AutoInt}_{\text{text}}$, and $\text{GDCN}_{\text{text}}$.

**Group C: Deep CTR models with *semantic embedding*.** Here, item embeddings are initialized with pretrained semantic representations:

$$\mathbf{t} = g\left(\mathbf{w}_t\right), \tag{5}$$

where $g$ represents a large language model. The models selected for benchmarking are $\text{DNN}_{\text{emb}}$, $\text{DCNv2}_{\text{emb}}$, $\text{AutoInt}_{\text{emb}}$, and $\text{GDCN}_{\text{emb}}$ models.

**Group D: LLM with *unique identifier*.** Following P5 [48], we treat item unique identifiers as special tokens and fine-tune LLMs for recommendation. The classification logits $l_{\text{yes}}$ and $l_{\text{no}}$ for the YES and NO tokens are obtained from the final token. After softmax normalization over these two tokens, the click probability is:

$$\hat{y}_{u,t} = \frac{e^{l_{\text{yes}}}}{e^{l_{\text{yes}}} + e^{l_{\text{no}}}}. \tag{6}$$

The models selected for benchmarking are P5-BERT$_{\text{base}}$, P5-OPT$_{\text{350M}}$, P5-OPT$_{\text{1B}}$, and P5-Llama-3$_{\text{7B}}$.

**Group E: LLM with *text*.** Here, items are represented solely by their textual features, without adding extra tokens. Owing to their natural language understanding, these LLMs are evaluated in both zero-shot and fine-tuned settings. We benchmark general-purpose models such as GPT-3.5 [45], the LLaMA series [60, 61, 16], Qwen [62], OPT [63], Phi [64], Mistral [65], GLM [66], DeepSeek-Qwen-2 [67], as well as recommendation-specific models like P5 [48] and RecGPT [17].

**Group F: LLM with *semantic identifier*.** For this group, we replace single unique identifier with multiple semantic identifiers per item. The models selected for benchmarking are SID-BERT$_{\text{base}}$ and SID-OPT$_{\text{350M}}$, which use BERT$_{\text{base}}$ and OPT$_{\text{350M}}$ as LLM backbone, respectively.

## 3.2 List-wise Recommendation

List-wise recommendation predicts the next item $t_{u_x}$ that a user $u$ will interact with, given their historical behavior sequence $\mathbf{s}_u = \{s_{u_i}\}_{i=1}^{x-1}$. The model is trained with categorical cross-entropy loss:

$$\mathcal{L} = -\sum_{u \in \mathcal{U}} \log \frac{\exp\left(f(\mathbf{s}_u, t_{u_x})\right)}{\sum_{t' \in \mathcal{T}} \exp\left(f(\mathbf{s}_u, t')\right)}, \tag{7}$$

where $\mathcal{U}$ denotes the set of users, $t_{u_x}$ is the true next item, $\mathcal{T}$ is the candidate set, and $f(\mathbf{s}_u, t')$ computes the compatibility score.

**Group G: SeqRec models with *unique identifier*.** We benchmark a typical sequential recommendation model, SASRec [43], which uses unique item identifiers. The prediction score is computed as:

$$f(\mathbf{s}_u, t_{u_i}) = \mathbf{v}_{u_i}^T \mathbf{h}_{u_{i-1}}, \tag{8}$$

where $\mathbf{h}_{u_{i-1}}$ contains user history up to $i-1$, and $\mathbf{v}_{u_i}$ is the latent classification vector for item $t_{u_i}$.

**Group H: SeqRec models with *semantic identifier*.** In this group, we extend the next-token prediction task (as in Group G) by representing each item with multiple semantic identifiers instead of a single unique identifier. This formulation decomposes an item into a sequence of tokens, where each valid token combination corresponds to a specific item. We benchmark the SID-SASRec model, which use SASRec model as backbone. During inference, we employ an autoregressive decoding strategy using beam search. At each decoding step, the model predicts a set of candidate tokens and maintains the top K partial sequences (beams) based on their cumulative scores. However, since the item representation is structured as a path in a pre-constructed *semantic identifier* tree, standard beam search can produce token sequences that do not correspond to any valid item.

To overcome this limitation, we introduce a conditional beam search (CBS) technique. In our CBS approach, the *semantic identifier* tree organizes valid token sequences as paths from the root to a leaf node. At every decoding step, the candidate tokens for each beam are filtered to retain only those that extend the current partial sequence to a valid prefix in the *semantic identifier* tree. This restriction ensures that each beam can eventually form a complete, valid item identifier. Only the tokens that lead to a leaf node–representing a complete and valid *semantic identifier* sequence–are allowed to contribute a positive prediction logit. We use $_{\text{-CBS}}$ to denote inference with CBS.

**Group I: LLMs with *unique identifier*.** We extend the LLM-based framework to list-wise recommendation by incorporating item unique identifiers directly into the input prompt. The model is fine-tuned on the next-item prediction task by minimizing the categorical cross-entropy loss (Eqs. 7 and 8). P5-BERT$_{\text{base}}$, P5-Qwen-2$_{0.5B}$, P5-Qwen-2$_{1.5B}$, and P5-Llama$_{7B}$ are chosen for benchmarking.

**Group J: LLMs with *semantic identifier*.** Compared with Group I, we replace item unique identifier with semantic identifier in the input prompt. The model is fine-tuned using the categorical cross-entropy loss as in Eqs. 7 and 8. Conditional beam search is employed to ensure the decoded *semantic identifier* sequence maps to a valid item. We benchmark SID-BERT$_{\text{base}}$ and SID-Llama-3$_{7B}$.

## 4 Experimental Settings

### 4.1 Datasets

To avoid reliance on a single platform, we conduct all the experiments on five datasets from distinct domains and institutions: **H&M** for fashion recommendation, **MIND** for news recommendation, **MicroLens** for video recommendation, **Goodreads** for book recommendation, and **CDs** for music recommendation. Moreover, since the training and testing data sizes of the original datasets vary significantly, the comprehensive evaluation scores of the final models could be influenced by these discrepancies. To mitigate this issue, we perform uniform preprocessing on all datasets to obtain approximately similar dataset sizes. The specific details of the datasets are summarized in Table 4.

### 4.2 Evaluation Metrics

Following common practice [68, 69], we evaluate recommendation performance using widely adopted metrics, including ranking metrics such as GAUC, NDCG, and MRR, as well as matching metrics

like F1 and RECALL. However, due to space limitations, we present only the GAUC metric for pair-wise recommendation tasks and NDCG@10 for list-wise recommendation scenarios. The full evaluation results will available on our webpage.

All the experiments are trained on a single Nvidia A100 GPU device. Except for the zero-shot setting, all results are averaged over five runs, with statistically significant differences observed ($p < 0.05$). Moreover, we use the latency (ms) metric to evaluate the model's inference efficiency, calculated as the average time per inference over 1,000 runs.

### 4.3 Implementation Details

**Data Pre-processing.** For datasets lacking user behavior sequences (i.e., H&M, CDs, Goodreads, and MicroLens), we construct these sequences by arranging each user's positive interactions in chronological order. In the pair-wise recommendation scenario, for datasets without provided negative samples (i.e., MicroLens and H&M), we perform negative sampling for each user with a negative ratio of 2. Additionally, we truncate user behavior sequences to a maximum length of 20 to ensure consistency across datasets. For deep CTR models, i) we utilize the nltk package to tokenize the text data and subsequently retain only those tokens present in the GloVe vocabulary [70] under the *text* settings, and we did not use pretrained GloVe vectors during training; ii) we use Llama-1$_{7B}$ model to extract the pretrained item embeddings under the *semantic embedding* settings.

**Semantic Identifier Generation.** We employ the pipeline proposed by TIGER [32] to generate *semantic identifier*. First, we use an LLM, i.e., SentenceBERT [71], to extract embeddings for each item content. Then, we perform discretization training using the RQ-VAE [30] model on these embeddings. Following common practice [32, 31, 33], we utilize a 4-layer codebook, with each layer having a size of 256. The representation space of this codebook approximately reaches 4 billion.

**Identifier Vocabulary.** Regardless of whether we use *unique identifier* or *semantic identifier*, we construct new identifier vocabularies for the LLM. Specifically, the vocabulary size $V$ matches the number of items when using *unique identifier*, or $V = 256 \times 4 = 1,024$ when using *semantic identifier*. We initialize a randomly generated embedding matrix $\mathbf{E}_{id} \in \mathbb{R}^{V \times d}$, where $d$ is the embedding dimension of the current LLM.

**Model Fine-tuning.** We employ the low-rank adaptation (LoRA) technique [72] for parameter-efficient fine-tuning of large language models. For the pair-wise recommendation scenario, LoRA is configured with a rank of 32 and an alpha of 128, whereas for the list-wise recommendation scenario, these parameters are set to (128, 128). The learning rate is fixed at $1 \times 10^{-4}$ for LLM-based models and $1 \times 10^{-3}$ for other models. In addition, we set the batch size to 5,000 for all deep CTR models, 64 for models with fewer than 7B parameters, and 16 for models with 7B parameters. All the experiments are conducted on a single Nvidia A100 GPU device. Except for the zero-shot setting, all results are averaged over five runs, with statistically significant differences observed (p ¡ 0.05).

## 5 Pair-wise Recommendation: Results and Findings

This section provides a detailed analysis of results from pair-wise recommendation.[3]

### 5.1 Zero-shot vs. Fine-tuned LLMs

> Most LLMs exhibit limited zero-shot recommendation capabilities; however, models pre-trained on data with implicit recommendation signals, such as Mistral [65], GLM [66], and Qwen-2 [62], perform significantly better. Fine-tuning boosts the recommendation accuracy of LLMs, with Llama-3$_{7B}$ improving by up to 43%.

Table 2 (last block) presents the zero-shot performance of various LLMs on pair-wise recommendation scenario. Our findings reveal that most LLMs struggle with general recommendation tasks. These models appear to have difficulty extracting user interest patterns from behavior sequences and

---

[3]Due to space limits, additional results are in the appendix and supplement.

Table 2: Comparison between fine-tuned LLM and conventional DLRMs in the **pair-wise** recommendation scenario. We report the AUC metric. CPU and GPU inference time are in millisecond.

| Item Representation | Recommender | 📑 MIND | 🎬 MicroLens | 📖 Goodreads | 💿 CDs | 🛍 H&M | Overall | CPU | GPU |
|---|---|---|---|---|---|---|---|---|---|
| **Conventional DLRMs** | | | | | | | | | |
| *unique identifier* | DNN | 0.6692 | 0.7421 | 0.5831 | 0.5757 | 0.7952 | 0.6731 | 0.43 | 1.69 |
| | PNN | 0.6581 | 0.7359 | 0.5801 | 0.5331 | 0.7648 | 0.6544 | 1.00 | 1.70 |
| | DeepFM | 0.6670 | 0.7594 | 0.5782 | 0.5681 | 0.7749 | 0.6695 | 0.98 | 1.65 |
| | DCN | 0.6625 | 0.7410 | 0.5902 | 0.5780 | 0.7913 | 0.6726 | 1.07 | 1.67 |
| | DCNv2 | 0.6707 | 0.7578 | 0.5778 | 0.5664 | 0.7950 | 0.6735 | 4.29 | 3.62 |
| | MaskNet | 0.6631 | 0.7179 | 0.5719 | 0.5532 | 0.7481 | 0.6508 | 3.08 | 1.71 |
| | FinalMLP | 0.6649 | 0.7600 | 0.5807 | 0.5670 | 0.7858 | 0.6717 | 1.72 | 1.74 |
| | AutoInt | 0.6690 | 0.7451 | 0.5879 | 0.5789 | 0.8027 | 0.6767 | 1.42 | 2.29 |
| | GDCN | 0.6704 | 0.7571 | 0.5948 | 0.5784 | 0.8120 | 0.6825 | 1.20 | 2.02 |
| *text* | $DNN_{text}$ | 0.6867 | 0.7741 | 0.5857 | 0.5655 | 0.8475 | 0.6919 | 4.73 | 3.72 |
| | $DCNv2_{text}$ | 0.6802 | 0.7804 | 0.5789 | 0.5577 | 0.8560 | 0.6906 | 4.91 | 3.12 |
| | $AutoInt_{text}$ | 0.6701 | 0.7761 | 0.5803 | 0.5687 | 0.8490 | 0.6888 | 5.39 | 3.87 |
| | $GDCN_{text}$ | 0.6783 | 0.7842 | 0.5796 | 0.5641 | 0.8555 | 0.6923 | 5.09 | 3.77 |
| *semantic embedding* | $DNN_{emb}$ | 0.7154 | 0.8141 | 0.5997 | 0.5848 | 0.8717 | 0.7171 | 1.42 | 2.09 |
| | $DCNv2_{emb}$ | 0.7167 | 0.8061 | 0.5999 | 0.5944 | 0.8626 | 0.7159 | 8.28 | 5.31 |
| | $AutoInt_{emb}$ | 0.7081 | 0.8099 | 0.6015 | 0.5560 | 0.8594 | 0.7070 | 2.04 | 3.09 |
| | $GDCN_{emb}$ | 0.7093 | 0.7997 | 0.5943 | 0.5828 | 0.8565 | 0.7085 | 1.77 | 2.54 |
| **Fine-tuned LLMs** | | | | | | | | | |
| *unique identifier* | $P5\text{-}BERT_{base}$ | 0.5507 | 0.5850 | 0.5038 | 0.5162 | 0.5402 | 0.5392 | 36.40 | 8.33 |
| | $P5\text{-}OPT_{base}$ | 0.6330 | 0.5099 | 0.5031 | 0.4989 | 0.4939 | 0.5278 | 286.56 | 16.37 |
| | $P5\text{-}OPT_{large}$ | 0.6512 | 0.6984 | 0.5110 | 0.5281 | 0.6177 | 0.6013 | 950.89 | 15.50 |
| | $P5\text{-}Llama\text{-}3_{7B}$ | 0.6697 | 0.7457 | 0.5780 | 0.5688 | 0.7260 | 0.6576 | 6350 | 35.20 |
| *text* | $BERT_{base}$ | 0.7175 | 0.8066 | 0.5148 | 0.5789 | 0.8635 | 0.6962 | 53.26 | 11.08 |
| | $OPT_{1B}$ | 0.7346 | 0.8016 | 0.5889 | 0.5850 | 0.5121 | 0.6444 | 1140 | 16.38 |
| | $Llama\text{-}3_{7B}$ | 0.7345 | **0.8328** | **0.6826** | 0.6268 | 0.8771 | 0.7508 | 6800 | 65.11 |
| | $Mistral\text{-}2_{7B}$ | **0.7353** | 0.8295 | 0.6680 | **0.6754** | **0.8810** | **0.7578** | 7680 | 76.14 |
| *semantic identifier* | $SID\text{-}BERT_{base}$ | 0.5704 | 0.5860 | 0.4914 | 0.5042 | 0.5401 | 0.5384 | 3640 | 8.33 |
| | $SID\text{-}OPT_{base}$ | 0.5987 | 0.4989 | 0.5004 | 0.4977 | 0.4957 | 0.5183 | 286.56 | 16.37 |
| **Zero-shot LLMs** | | | | | | | | | |
| *text* | $BERT_{base}$ | 0.4963 | 0.4992 | 0.4958 | 0.5059 | 0.5204 | 0.5035 | 53.26 | 11.08 |
| | $OPT_{350M}$ | 0.5490 | 0.4773 | 0.5015 | 0.5093 | 0.4555 | 0.4985 | 332.34 | 14.99 |
| | $OPT_{1B}$ | 0.5338 | 0.5236 | 0.5042 | 0.4994 | 0.5650 | 0.5252 | 1140 | 16.38 |
| | $Llama\text{-}1_{7B}$ | 0.4583 | 0.4572 | 0.4994 | 0.4995 | 0.4035 | 0.4636 | 3170 | 71.10 |
| | $Llama\text{-}2_{7B}$ | 0.4945 | 0.4877 | 0.5273 | 0.5191 | 0.4519 | 0.4961 | 6200 | 71.90 |
| | $Llama\text{-}3_{8B}$ | 0.4904 | 0.5577 | 0.5191 | 0.5136 | 0.5454 | 0.5252 | 6800 | 65.11 |
| | $Llama\text{-}3.1_{8B}$ | 0.5002 | 0.5403 | 0.5271 | 0.5088 | 0.5462 | 0.5245 | 6580 | 66.39 |
| | $Mistral_{7B}$ | 0.6300 | 0.6579 | **0.5718** | 0.5230 | 0.7166 | 0.6199 | 7680 | 76.14 |
| | $GLM\text{-}4_{9B}$ | **0.6304** | **0.6647** | 0.5671 | 0.5213 | **0.7319** | **0.6231** | 9690 | 83.38 |
| | $Qwen\text{-}2_{0.5B}$ | 0.4868 | 0.5717 | 0.5148 | 0.5043 | 0.6287 | 0.5413 | 543.73 | 34.89 |
| | $Qwen\text{-}2_{1.5B}$ | 0.5411 | 0.6072 | 0.5264 | 0.5174 | 0.6615 | 0.5707 | 1420 | 40.01 |
| | $Qwen\text{-}2_{7B}$ | 0.5862 | 0.6640 | 0.5494 | **0.5256** | 0.7124 | 0.6075 | 6150 | 70.47 |
| | $DS\text{-}Qwen\text{-}2_{7B}$ | 0.5127 | 0.5631 | 0.5165 | 0.5146 | 0.5994 | 0.5413 | 7520 | 61.60 |
| | $Phi\text{-}2_{3B}$ | 0.4851 | 0.5078 | 0.5049 | 0.4991 | 0.5447 | 0.5083 | 2100 | 61.58 |
| | GPT-3.5 | 0.5057 | 0.5110 | 0.5122 | 0.5046 | 0.5801 | 0.5227 | - | - |
| | $RecGPT_{7B}$ | 0.5078 | 0.4703 | 0.5083 | 0.5019 | 0.4875 | 0.4952 | 7160 | 54.34 |
| | $P5_{Beauty}$ | 0.4911 | 0.5017 | 0.5027 | 0.5447 | 0.4845 | 0.5049 | 74.11 | 12.30 |

assessing the relevance between user interests and candidate items. Moreover, specialized recommendation models such as P5 [48] and RecGPT [17] also underperformed in our evaluations. They can effectively capture item semantics on fine-tuned datasets but lack strong generalization and zero-shot inference capabilities. Notably, the Mistral [65], GLM [66], and Qwen-2 [62] models demonstrated comparatively robust CTR prediction performance, with the recommendation effectiveness of Qwen-2 showing a positive correlation with model size. These models may have been exposed to diverse web content, including user interactions signals, enhancing their generalization for recommendation tasks.

We perform instruction tuning on various LLMs to align them with recommendation tasks. Table 2 shows that this fine-tuning yields a relative improvement in recommendation accuracy ranging from 22% to 43%, highlighting the importance of domain-specific alignment. Notably, $Llama\text{-}3_{7B}$ outperformed $Mistral\text{-}2_{7B}$ on the MicroLens and Goodreads datasets. Although Mistral-2 ranked in the top three for zero-shot scenarios, Llama-3's overall performance of was comparable, while smaller models such as BERT and OPT consistently lagged behind. These results emphasize the superior semantic understanding and deep reasoning capabilities inherent in larger models.

Table 3: Comparison between LLM recommenders and conventional DLRMs in the **list-wise** recommendation scenario. We display NDCG@10 metric in this table. CPU inference time are in millisecond. "$_{\text{-CBS}}$" means using conditional beam search (see Sec 3.2) during inference.

| Item Representation | Recommender | 📰 MIND | 📺 MicroLens | 📖 Goodreads | 💿 CDs | 👜 H&M | Overall | CPU |
|---|---|---|---|---|---|---|---|---|
| **Conventional DLRMs** | | | | | | | | |
| *unique identifier* | SASRec$_{3L}$ | 0.0090 | 0.0000 | 0.0165 | 0.0016 | 0.0209 | 0.0096 | 23.30 |
| | SASRec$_{6L}$ | 0.0097 | 0.0006 | 0.0224 | 0.0012 | 0.0297 | 0.0127 | 38.43 |
| | SASRec$_{12L}$ | 0.0241 | 0.0297 | 0.0548 | 0.1041 | 0.1235 | 0.0672 | 51.77 |
| | SASRec$_{24L}$ | 0.0119 | 0.0312 | 0.0601 | 0.1267 | 0.1191 | 0.0698 | 103.41 |
| *semantic identifier* | SID-SASRec$_{3L}$ | 0.0266 | 0.0028 | 0.0029 | 0.0000 | 0.0084 | 0.0081 | 36.12 |
| | SID-SASRec$_{3L\text{-CBS}}$ | 0.0849 | 0.0123 | 0.0127 | 0.0007 | 0.0422 | 0.0306 | 66.67 |
| | SID-SASRec$_{6L}$ | 0.0225 | 0.0047 | 0.0038 | 0.0140 | 0.0097 | 0.0109 | 59.08 |
| | SID-SASRec$_{6L\text{-CBS}}$ | 0.0647 | 0.0179 | 0.0141 | 0.0331 | 0.0406 | 0.0341 | 90.41 |
| | SID-SASRec$_{12L}$ | 0.0201 | 0.0044 | 0.0039 | 0.0136 | 0.0165 | 0.0117 | 1310 |
| | SID-SASRec$_{12L\text{-CBS}}$ | 0.0695 | 0.0234 | 0.0140 | 0.0324 | 0.0598 | 0.0398 | 1340 |
| **Fine-tuned LLMs** | | | | | | | | |
| *unique identifier* | P5-BERT$_{base}$ | 0.0430 | **0.1867** | **0.0557** | 0.1198 | 0.1075 | 0.1025 | 41.54 |
| | P5-QWen-2$_{0.5B}$ | 0.0549 | 0.0201 | 0.0322 | 0.0128 | 0.0234 | 0.0287 | 556.95 |
| | P5-QWen-2$_{1.5B}$ | 0.0506 | 0.0254 | 0.0316 | 0.0015 | 0.0217 | 0.0262 | 1120 |
| | P5-Llama-3$_{7B}$ | 0.0550 | 0.0178 | 0.0134 | 0.0072 | 0.0353 | 0.0257 | 28060 |
| *semantic identifier* | SID-BERT$_{base}$ | 0.0654 | 0.0022 | 0.0025 | 0.3539 | 0.0467 | 0.0941 | 1830 |
| | SID-BERT$_{base\text{-CBS}}$ | **0.1682** | 0.1195 | 0.0059 | 0.4616 | **0.1834** | **0.1877** | 1900 |
| | SID-Llama-3$_{7B}$ | 0.0456 | 0.0255 | 0.0221 | 0.2443 | 0.0337 | 0.0742 | 167250 |
| | SID-Llama-3$_{7B\text{-CBS}}$ | 0.1677 | 0.0827 | 0.0508 | 0.3898 | 0.1125 | 0.1607 | 177540 |

## 5.2 Performance Comparison: LLMs vs. Conventional Deep CTR Models

> Large-scale LLMs (e.g., Llama, Mistral) achieve over a 5% improvement in recommendation accuracy compared to the best conventional recommender (DNN$_{\text{emb}}$) using *semantic embedding*. However, these gains come with significant latency; the best conventional recommender retains 95% of the performance while operating thousands of times faster.

Table 2 compares recommendation performance using various item representation forms for both conventional recommenders (i.e., DLRM) and LLM-based approaches. The key findings are:

**Firstly,** even without textual modalities, conventional *unique identifier*-based CTR models outperform the zero-shot LLM-based recommenders, highlighting the importance of interaction data. Moreover, fine-tuned *unique identifier*-based LLMs still lag behind, likely because they struggle to capture explicit feature interactions. **Secondly,** incorporating textual data into CTR models yields significant gains. We did not use pretrained word embeddings, as the item-side text itself effectively learns robust item relationships. **Thirdly,** initializing item representations with embeddings from Llama-1 for *semantic embedding*-based CTR models introduces high-quality semantic information, outperforming both prior methods and small *text*-based LLMs (e.g., BERT, OPT) due to Llama's superior semantic quality and deeper network architecture. **Fourthly,** *text*-based LLMs using large models like Llama-3 and Mistral-2 outperform all baselines, demonstrating their disruptive potential in recommendation tasks. **Fifthly,** conversely, fine-tuning *semantic identifier*-based LLMs yields poor performance in CTR scenarios, likely due to smaller models' limited ability to learn discrete semantic information. **Sixthly,** in terms of efficiency, *semantic embedding*-based CTR models within the LLM-FOR-RS paradigm offer the best cost-effectiveness with minimal modifications to traditional architectures, making this approach one of the most practical in industry.

# 6 List-wise Recommendation: Results and Findings

This section presents results from list-wise recommendation. Sequential recommenders [43, 32] usually rely on next-item prediction, which doesn't align with using *text* as item representation. Thus, we evaluate two forms: *unique identifier* and *semantic identifier*. Since LLMs are unable to recognize unseen tokens, they lack zero-shot recommendation abilities and require fine-tuning.

## 6.1 Unique ID vs. Semantic ID

> Overall, *semantic identifier* has shown to be a more effective representation than *unique identifier*, whether integrated with LLMs or traditional recommenders, highlighting the value of incorporating item content knowledge into sequential recommenders.

Table 3 evaluates the recommendation abilities of LLMs and conventional DLRMs in the list-wise recommendation scenario, leading to the following observations: **Firstly,** within the SASRec series, performance generally improves with more transformer layers, reflecting the scaling behavior of conventional sequential recommenders. Notably, SID-SASRec outperforms standard SASRec with fewer layers, suggesting that *semantic identifier*, by decomposing item representations into logically and hierarchically structured tokens, allows shallower networks to better capture user interests. However, as layers increases, *semantic identifier*'s advantage diminishes, likely because deeper SASRec architectures can more effectively learn user sequence patterns, even without pretrained semantic information. **Secondly,** comparing pairs like (P5-BERT$_{base}$, SID-BERT$_{base\text{-}CBS}$) and (P5-Llama-3$_{8B}$, SID-Llama-3$_{7B\text{-}CBS}$), LLMs with *semantic identifier* consistently outperform their *unique identifier* counterparts, with improvements up to 83%. This highlights the efficiency and potential of *semantic identifier* representation in enhancing recommendation performance.

## 6.2 Performance Comparison: LLMs vs. Conventional Sequential Recommenders

> LLMs outperform traditional sequential recommenders in accuracy using either *unique identifier* or *semantic identifier* representations, but their inference efficiency remains a critical issue requiring urgent improvement.

Based on *unique identifier* representations, the BERT$_{base}$ model outperforms both SASRec$_{12L}$—which shares the same network architecture as BERT$_{base}$—and the deeper SASRec$_{24L}$. Despite the absence of textual features in item representations, this observation suggests that language patterns acquired during pretraining bear an abstract similarity to user interest patterns in recommender systems, thereby facilitating effective knowledge transfer. Furthermore, LLM recommenders employing *semantic identifier* representations exhibit markedly superior performance compared to the SID-SASRec series. By incorporating semantic item knowledge, *semantic identifier* enables LLMs to more effectively interpret user sequences and capture high-quality user interests. Additionally, models utilizing conditional beam search constraints (the $_{\text{-}CBS}$ series) achieve further improvements in recommendation performance. However, these gains come at a substantial cost in inference efficiency; overall, LLM recommenders require nearly 1,000 times more inference time than SASRec. This significant efficiency gap represents a critical challenge that should be addressed to ensure the practical deployment of LLM recommenders.

## 7 Conclusion

We introduced RECBENCH, a comprehensive benchmark for comparing LLM-AS-RS and DLRMs. Our study systematically explores various item representation forms and covers both click-through rate prediction and sequential recommendation tasks across diverse datasets and models. While LLM-based recommenders, particularly those using large-scale models, achieve significant performance gains, they face substantial efficiency challenges compared to conventional DLRMs. This trade-off highlights the need for research into inference acceleration techniques, crucial for deploying LLM-based recommenders in high-throughput industrial settings.

## Acknowledgments

We sincerely thank the anonymous reviewers for their valuable feedback. This research was supported in part by the Collaborative Research Project P0056067, funded by Huawei International Pte. Ltd.

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

# A  Limitations

This study evaluates the recommendation capabilities of large language models and traditional recommender systems across two distinct scenarios: pair-wise, similar to click-through rate (CTR) models, and list-wise, akin to sequential recommenders. Our current analysis does not address fairness and privacy considerations, which are essential areas for future improvements of RECBENCH. Due to space limits, we present only the AUC metric in the main text. However, additional metrics, including MRR and nDCG, which demonstrate high consistency with the AUC results, are included in the supplementary materials.

# B  Broader Impacts

Our benchmark offers a comprehensive, scalable framework for evaluating foundation models in recommendation scenarios, fostering systematic and reproducible research. By encompassing diverse domains and datasets, it assesses the generalization capabilities of large models. Moreover, its openness and reusability reduce experimental costs, lowering the entry barrier for both academic and industrial practitioners. However, since foundation models can amplify data biases or user stereotypes, we advise users to exercise caution and conduct ethical audits when applying these models in real-world systems.

Table 4: Datasets statistics.

| Dataset | | H&M | MIND | MicroLens | Goodreads | CDs |
|---|---|---|---|---|---|---|
| **Type** | | Fashion | News | Video | Book | Music |
| **Text Attribute** | | desc | title | title | name | name |
| **Pair-wise Test set** | #Sample | 20,000 | 20,006 | 20,000 | 20,009 | 20,003 |
| | #Item | 26,270 | 3,088 | 15,166 | 26,664 | 36,765 |
| | #User | 5,000 | 1,514 | 5,000 | 1,736 | 4,930 |
| **Pair-wise Finetune set** | #Sample | 100,000 | 100,000 | 100,000 | 100,005 | 100,003 |
| | #Item | 60,589 | 17,356 | 19,111 | 74,112 | 113,671 |
| | #User | 25,000 | 8,706 | 25,000 | 8,604 | 24,618 |
| **List-wise Test set** | #Seq | 5,000 | 5,000 | 5,000 | 5,000 | 5,000 |
| | #Item | 15,889 | 10,634 | 12,273 | 38,868 | 19,684 |
| **List-wise Finetune set** | #Seq | 40,000 | 40,000 | 40,000 | 40,000 | 40,000 |
| | #Item | 35,344 | 24,451 | 18,841 | 136,296 | 95,409 |

Table 5: Additional evaluation metrics for the **pair-wise** recommendation scenario.

| Item Representation | Recommender | MIND | | | H&M | | |
|---|---|---|---|---|---|---|---|
| | | AUC | nDCG@1 | nDCG@5 | AUC | nDCG@1 | nDCG@5 |
| **Conventional DLRMs** | | | | | | | |
| *text* | DNN | 0.6867 | 0.5324 | 0.5831 | 0.8475 | 0.8560 | 0.9337 |
| | DCNv2 | 0.6802 | 0.5357 | 0.5820 | 0.8560 | 0.8662 | 0.9376 |
| | AutoInt | 0.6701 | 0.5277 | 0.5730 | 0.8490 | 0.8622 | 0.9350 |
| | GDCN | 0.6783 | 0.5257 | 0.5751 | 0.8555 | 0.8658 | 0.9374 |
| *semantic embedding* | DNN | 0.7154 | 0.5618 | 0.6226 | 0.8717 | 0.8972 | 0.9472 |
| | DCNv2 | 0.7167 | 0.5627 | 0.6272 | 0.8626 | 0.8848 | 0.9427 |
| | AutoInt | 0.7081 | 0.5561 | 0.6123 | 0.8594 | 0.8792 | 0.9408 |
| | GDCN | 0.7093 | 0.5677 | 0.6176 | 0.8565 | 0.8782 | 0.9399 |
| **Zero-shot LLMs** | | | | | | | |
| *text* | BERT$_{base}$ | 0.4963 | 0.2655 | 0.3729 | 0.5204 | 0.4934 | 0.7899 |
| | OPT$_{1B}$ | 0.5338 | 0.2981 | 0.4077 | 0.5650 | 0.5302 | 0.8060 |
| | Llama-1$_{7B}$ | 0.4583 | 0.2100 | 0.3301 | 0.4035 | 0.3706 | 0.7391 |
| | Llama-3.1$_{8B}$ | 0.5002 | 0.3007 | 0.3907 | 0.5462 | 0.5362 | 0.8028 |
| **Fine-tune LLMs** | | | | | | | |
| *text* | BERT$_{base}$ | 0.7175 | 0.5862 | 0.6274 | 0.8635 | 0.8812 | 0.9423 |
| | OPT$_{1B}$ | 0.7346 | 0.6042 | 0.6488 | 0.5121 | 0.5129 | 0.7904 |
| | Llama-3.1$_{8B}$ | 0.7345 | 0.6040 | 0.6490 | 0.8771 | 0.8917 | 0.9476 |
| | Mistral$_{7B}$ | 0.7353 | 0.6057 | 0.6464 | 0.8810 | 0.9021 | 0.9504 |

# C    Technical Appendices

## C.1    Additional Evaluation Metrics

Due to space constraints, Table 2 and Table 3 report only AUC and nDCG@10, respectively. For a more comprehensive comparison, we present results in additional evaluation metrics.

As shown in Table 5, the nDCG results are generally consistent with the AUC scores. Similarly, Table 6 shows that MRR and Recall align well with nDCG@10. These findings underscore the robustness of our conclusions. Full experimental results will be made available on our website.

Table 6: Additional evaluation metrics for the **list-wise** recommendation scenario. Experiments are conducted on the MIND dataset.

| Item Representation | Recommender | nDCG@1 | nDCG@5 | nDCG@10 | MRR | Recall@5 | Recall@10 |
|---|---|---|---|---|---|---|---|
| **Conventional DLRMs** | | | | | | | |
| *unique identifier* | $SASRec_{3L}$ | 0.0066 | 0.0086 | 0.0090 | 0.0083 | 0.0102 | 0.0114 |
| | $SASRec_{6L}$ | 0.0074 | 0.0091 | 0.0097 | 0.0090 | 0.0108 | 0.0126 |
| *semantic identifier* | $SASRec_{3L}$ | 0.0124 | 0.0215 | 0.0266 | 0.0220 | 0.0298 | 0.0464 |
| | $SASRec_{3L-CBS}$ | 0.0792 | 0.0847 | 0.0849 | 0.0836 | 0.0882 | 0.0888 |
| | $SASRec_{6L}$ | 0.0108 | 0.0209 | 0.0225 | 0.0194 | 0.0294 | 0.0342 |
| | $SASRec_{6L}$-CBS | 0.0608 | 0.0644 | 0.0647 | 0.0636 | 0.0670 | 0.0680 |
| **Fine-tuned LLMs** | | | | | | | |
| *unique identifier* | $P5-BERT_{base}$ | 0.0138 | 0.0290 | 0.0430 | 0.0302 | 0.0532 | 0.0985 |
| | $P5-Qwen-2_{0.5B}$ | 0.0152 | 0.0394 | 0.0549 | 0.0409 | 0.0646 | 0.1132 |
| | $P5-Qwen-2_{1.5B}$ | 0.0148 | 0.0353 | 0.0506 | 0.0387 | 0.0574 | 0.1052 |
| | $P5-Llama-3_{8B}$ | 0.0156 | 0.0393 | 0.0550 | 0.0421 | 0.0638 | 0.1126 |
| *semantic identifier* | $SID-BERT_{base}$ | 0.0412 | 0.0641 | 0.0654 | 0.0581 | 0.0842 | 0.0884 |
| | $SID-BERT_{base-CBS}$ | 0.1384 | 0.1604 | 0.1682 | 0.1577 | 0.1806 | 0.2048 |
| | $SID-Llama-3_{8B}$ | 0.0152 | 0.0356 | 0.0456 | 0.0364 | 0.0552 | 0.0862 |
| | $SID-Llama-3_{8B-CBS}$ | 0.1670 | 0.1676 | 0.1677 | 0.1675 | 0.1680 | 0.1682 |

Table 7: Extended results of conventional DLRMs for the **list-wise** recommendation scenario: Supplementary to Table 3.

| Item Representation | Recommender | 📰 MIND | 📺 MicroLens | 📖 Goodreads | 🧭 CDs | 🧱 H&M |
|---|---|---|---|---|---|---|
| *unique identifier* | GRU4Rec | 0.0040 | 0.0000 | 0.0073 | 0.0015 | 0.0237 |
| | Caser | 0.0078 | 0.0000 | 0.0112 | 0.0014 | 0.0188 |
| | Bert4Rec | 0.0032 | 0.0000 | 0.0067 | 0.0014 | 0.0174 |
| | SASRec | 0.0090 | 0.0000 | 0.0165 | 0.0016 | 0.0209 |

## C.2 Results of Additional Conventional Sequential Recommenders

Due to space constraints, Table 3 includes only SASRec as a representative sequential recommender. In Table 7, we extend the comparison to additional conventional models, including GRU4Rec [73], Caser [74], and BERT4Rec [44], with the number of layers uniformly set to 3. SASRec consistently outperforms the other baselines, justifying its selection for more detailed comparisons with LLM-based recommenders (e.g., variations with *unique identifier*, *semantic identifier*, and different layer depths in Table 3).

## C.3 Prompt Analysis

We have provided the used prompt in our code repository. To further address your concern regarding the prompt templates, we conducted experiments using the prompts depicted in Table 8.

Specifically, P1 is a concise prompt; P2 is more detailed; and P2 (1-shot) follows the in-context learning paradigm by including a demonstration example. The average GAUC scores across five datasets (MIND, MicroLens, Goodreads, CDs, and H&M) using the $Qwen-3_{8B}$ model under these prompts are summarized in Table 9, from which we can observe that:

First, overall, for zero-shot performance, P2 offers only minor improvements over P1. Therefore, in our experiments, we used P1 for smaller models (size < 7B) with shorter input windows to conserve input tokens and allow for longer user sequences. For larger models with longer input windows, we used P2.

Second, the few-shot experiments were conducted three times, with each run using different examples randomly selected for in-context demonstrations. Compared to zero-shot performance, we observed that few-shot prompting does not always lead to improved results and can, in some cases, cause a significant drop in performance, as seen on the H&M dataset.

Table 8: List of prompts used in our benchmark.

| Version | Prompt |
|---------|--------|
| P1 (zero-shot) | You are a recommender. Please respond "YES" or "NO" to represent whether this user is interested in this item.
User history sequence: `[object Object]`.
Candidate item: `[object Object]`.
Answer (Yes/No): |
| P2 (zero-shot) | You are a recommender. I will provide user behavior sequence and a candidate item. Please respond "YES" or "NO" only. You are not allowed to give any explanation or note. Now, your role formally begins. Any other information should not disturb you.
User history sequence: `[object Object]`.
Candidate item: `[object Object]`.
Answer (Yes/No): |
| P2 (1-shot) | You are a recommender … (*example omitted*) …
User history sequence: `[object Object]`.
Candidate item: `[object Object]`.
Answer (Yes/No): |
| P2 (2-shot) | *Omitted due to space constraints.* |
| P2 (5-shot) | *Omitted due to space constraints.* |

Table 9: Impact of prompts. Experiments are conducted on the Qwen-$3_{8B}$ model.

| Setting | MIND | MicroLens | Goodreads | CDs | H&M |
|---------|------|-----------|-----------|-----|-----|
| P1 (zero-shot) | 0.5847 | 0.6543 | **0.5439** | 0.5180 | 0.6499 |
| P2 (zero-shot) | 0.6036 | **0.6618** | 0.5371 | 0.5175 | **0.6678** |
| P2 (1-shot) | 0.6097 (48) | 0.6254 (43) | 0.5356 (56) | 0.5056 (38) | 0.6178 (16) |
| P2 (2-shot) | **0.6354** (50) | 0.6536 (56) | 0.5360 (60) | 0.5150 (69) | 0.6105 (146) |
| P2 (5-shot) | 0.6208 (52) | 0.6565 (71) | 0.5429 (28) | **0.5188** (49) | 0.6176 (124) |

## C.4 Evaluation of More Recent Models

Here, we provide zero-shot and fine-tuning results (fine-tuned on CDs) using the Qwen3 series models.

The results in Table 10 demonstrate a clear trend consistent with scaling laws: model performance improves as model size increases, and fine-tuning yields significant performance gains. Overall, Qwen3-8B achieves zero-shot performance comparable to that of Qwen2-7B reported in the main paper.

Additionally, we intentionally omit models larger than 8B parameters. This decision is primarily motivated by the real-time constraints of recommendation systems, where low-latency inference is

Table 10: Evaluation on the Qwen-3 series.

| Dataset
Setting | MIND
Zero-shot | MicroLens
Zero-shot | Goodreads
Zero-shot | H&M
Zero-shot | CDs
Zero-shot | CDs
Fine-tune |
|--------|------|-----------|-----------|-----|-----|-----------|
| 0.6B | 0.4927 | 0.5165 | 0.5140 | 0.5993 | 0.5055 | 0.5098 |
| 1.7B | 0.5117 | 0.5704 | 0.5027 | 0.6571 | 0.4972 | 0.5775 |
| 4B | 0.5448 | 0.6394 | 0.5054 | 0.6487 | 0.5101 | 0.5879 |
| 8B | 0.6036 | 0.6618 | 0.5371 | 0.6678 | 0.5175 | 0.5919 |

Table 11: Impact of Dataset Size.

| Dataset | Train | Test | #Item | #User | Sparsity | Llama3-8B | DCNv2 | QWen3-8B |
|---------|-------|------|-------|-------|----------|-----------|-------|----------|
| **CDs-small** | 50,000 | 20,000 | 72,796 | 12,355 | 99.99444% | 0.6071 | 0.5463 | 0.5707 |
| **CDs** | 100,000 | 20,000 | 113,671 | 24,602 | 99.99642% | 0.6268 | 0.5664 | 0.5919 |
| **CDs-large** | 150,000 | 20,000 | 146,604 | 36,916 | 99.99723% | 0.6300 | 0.5714 | 0.6001 |

essential. In industrial settings, 8B already represents the practical upper limit for deployment in production environments, and model compression or distillation techniques are typically required to ensure efficiency. We will incorporate experiments on the Qwen3 series in the revised version.

## C.5 Impact of Dataset Characteristics

Here, we conducted an experiment where we varied the sparsity level through downsampling. We use a standard definition of sparsity, defined as:

$$sparsity = 1 - density = 1 - \frac{\#train\_sample}{\#items \times \#users}. \tag{9}$$

From Table 11, we can make the following observations:

As the number of training samples increases, data sparsity (as defined above) also increases. Despite this, the performance of Llama3-8B, Qwen3-8B, and DCNv2 consistently improves. This is because each user is represented by a sequence of item interactions, and with more item occurrences (i.e., more training samples), the item embeddings are trained more thoroughly and become more robust. Furthermore, the LLM-based models consistently outperform the DLRM (DCNv2) across all sparsity levels, demonstrating their effectiveness.

## C.6 Case Studies

Here, we present two case studies on the MIND dataset (for news recommendation) to illustrate how LLMs leverage semantic information in recommendation tasks. Each news article is represented by an ID beginning with "N" (e.g., "N50059") along with its article title.

**Case 1.** User click history: (1) N50059: This Ford GT40 Movie Rig From "Ford V Ferrari" Looks Absurd (2) N53017: Kendall Jenner Wore the Tiniest Dress to Go Jewelry Shopping

The positive candidate item, N35729: Porsche launches into second story of New Jersey building, killing 2, receives a high relevance score from LLMs but not from DLRMs. This demonstrates that LLMs can capture deeper semantic connections—such as the shared theme of luxury cars (Ferrari in N50059 and Porsche in N35729) – which are often overlooked by traditional ID-based CTR models.

**Case 2.** User click history: (1) N35: I Tried an Intense Metabolic Reset Program for a Month – and It Worked (2) N35452: Potentially historic wind event' over weekend could inflame California wildfires

The positive candidate item, N42634: The Latest Weight Loss Pills That Work And the Ones That Don't, does not share any overlapping words with N35. Despite this, LLMs assign it a high relevance score by recognizing its relevance to the user's interest in health and weight loss. This highlights the models' ability to understand and generalize semantic relationships, rather than relying only on direct word matches.

## C.7 Complexity Analysis on Conditional Beam Search

Based on the results in Table 3 of the main paper, we observe that CBS decoding is only marginally slower (approximately 3–5%) than standard beam search (BS) for large-size models. We provide the time complexity below:

In standard BS, the model maintains a set of $B$ partial hypotheses (beams), expanding each with all $V$ vocabulary tokens at each decoding step. At step $t$, for each beam, the model outputs a logit vector of size $V$, from which the top-$k$ candidates are selected based on cumulative scores. This process

Table 12: Fairness Comparison.

| Model | Inactive | Active | Overall | Diff. |
|---|---|---|---|---|
| Zero-shot | | | | |
| OPT-1B | 0.4996 | 0.5350 | 0.5011 | 0.0354 |
| Llama3-8B | 0.5206 | 0.5232 | 0.5207 | 0.0026 |
| Qwen3-4B | 0.5011 | 0.5273 | 0.5022 | 0.0262 |
| Qwen3-8B | 0.5153 | 0.5447 | 0.5163 | 0.0294 |
| Fine-tune | | | | |
| OPT-1B | 0.5790 | 0.6002 | 0.5799 | 0.0212 |
| Llama3-8B | 0.6184 | 0.6327 | 0.6190 | 0.0143 |
| Qwen3-4B | 0.5842 | 0.6042 | 0.5850 | 0.0200 |
| Qwen3-8B | 0.5994 | 0.6058 | 0.5997 | 0.0064 |
| DCNv2 | 0.5677 | 0.6071 | 0.5693 | 0.0394 |
| DCNv2-emb (w/ Llama3 embedding) | 0.5829 | 0.5886 | 0.5831 | 0.0057 |

is repeated for $T$ decoding steps. Therefore, the total time complexity of standard beam search is: $\mathcal{O}\left(B \cdot V \cdot T\right)$.

Under the same setup, CBS maintains full vocabulary logits but applies a dynamic mask over invalid tokens $\mathcal{O}\left(B \cdot V \cdot T\right)$, which matches the complexity of vanilla BS in asymptotic terms. Ideally, a more efficient implementation could reduce the complexity to $\mathcal{O}\left(B \cdot \bar{F} \cdot T\right)$, where $\bar{F}$ is the average number of valid next tokens per Trie node (typically $\bar{F} \ll V$). But in practice, we retain full-vocabulary logits to enable batched decoding, and thus the time complexity remains $\mathcal{O}\left(B \cdot V \cdot T\right)$.

Although CBS and standard beam search share the same theoretical time complexity, CBS is typically somewhat slower in practice. This slowdown is primarily due to additional operations such as dynamic logits masking and the overhead associated with Trie lookups. Nevertheless, given the performance gains relative to the modest increase in computational cost, the use of CBS remains well justified.

## C.8 Fairness and Privacy Issue

Fairness and privacy are critical considerations in real-world recommender systems, and we have proactively incorporated these principles into the design of RecBench:

- **User Split Strategy for Fairness**: To prevent unfair exposure bias toward highly active users, we ensure that each user appears exclusively in either the training or test set, but not both. This approach guarantees that all test users are unseen during training, reducing the risk of overfitting to popular users and promoting fairness in user representation.

- **Privacy-Preserving Design**: To protect user privacy, we do not use sensitive attributes such as user ID, location, or age during training. Instead, models are trained solely on behavioral history sequences, requiring them to learn and generalize user interest patterns without relying on private information. This approach not only safeguards privacy but also enhances model robustness.

we conducted additional experiments to evaluate model performance across user groups with varying activity levels. In our experimental setting, **all test users are unseen during training**; therefore, we define user activity according to the **length of each user's historical sequence in the test set**. Based on this definition, we establish a threshold as follows:

$$threshold = \frac{max\_length + min\_length}{2} \tag{10}$$

Table 13: Impact of Model Precision.

|  | Zero-shot | Fine-tune |
|---|---|---|
| BERT (float32) | 0.5204 | 0.8701 |
| BERT (bf16) | 0.5210 | 0.8688 |
| Llama3-8B (float32) | 0.5444 | 0.8598 |
| Llama3-8B (bf16) | 0.5454 | 0.8606 |

Users with sequence lengths below the threshold are labeled as *inactive*, while those above the threshold are labeled as *active*. All experiments are conducted on the Amazon CDs dataset. We compute GAUC for each group separately and report the difference as $Diff. = GAUC\_active - GAUC\_inactive$.

## C.9   Model Precision Analysis

We conducted experiments comparing 32- and 16-bit precision on the H&M dataset. The results (Table 13), reported using the GAUC metric, indicate that there are only minor differences in performance between the two precisions.

Therefore, to accelerate both training and inference in our benchmark, models larger than 7 billion parameters are fine-tuned using 16-bit precision, in line with common practice (e.g., LC-Rec [36] and GenRec [75], based on their official GitHub implementation).

