# OpenReview forum: "Can LLMs Outshine Conventional Recommenders? A Comparative Evaluation"
_NeurIPS.cc/2025/Datasets_and_Benchmarks_Track — NeurIPS 2025 Datasets and Benchmarks Track poster_

### Official Review · Reviewer_7Gns · 2025-06-05

**Rating:** 5
**Confidence:** 4

**Summary:**

This paper introduces a systematic benchmark comparing LLMs based recommenders against traditional deep learning recommenders.
It examines four forms of item representation, including unique identifier, text, semantic embedding, and semantic identifier, across CTR prediction and sequential recommendation.
Experiments involve 17 LLMs and ten conventional DLRMs, and report that fine-tuned LLM recommenders achieve better performances, but incur orders-of-magnitude higher inference latency.

**Dataset Code Accessibility:**

Yes

**Ethical Considerations:**

No, there are no or only very minor ethics concerns

**Final Justification:**

After the authors’ thorough responses to all reviews and the additional experiments, my endorsement of this paper has grown stronger.

**Limitations Weaknesses:**

1. Although the authors claim consistency, readers must probe supplementary materials to ensure robustness. Including at least one secondary metric in the main paper would facilitate quicker evaluation.
2. The efficiency section clearly shows that LLMs are far slower than DLRMs. However, there is no cost–benefit analysis, e.g., model‐size vs. incremental accuracy, that could guide practitioners in choosing a middle ground.

**Strengths Contributions:**

1. RECBENCH evaluates both CTR and SeqRec settings, covering zero-shot and fine-tuning regimes for LLMs.
2. The paper systematically explores four representation schemes: (a) unique identifier, (b) text, (c) semantic embedding, and (d) semantic identifier. This allows precise isolation of how each alignment strategy affects recommendation performance.
3. Benchmarked models include 10 conventional DLRMs and 17 LLMs spanning multiple institutions and sizes, ensures that reported findings are not artifacts of a particular model family.
4. 5 datasets from varied domains are used. This guards against overclaiming results that might hinge on a single data characteristic.
5. Beyond accuracy, RECBENCH measures inference latency per sample.

---

> ### Author Rebuttal · Authors · 2025-07-31
>
> Thank you for your positive feedback and recognition of RecBench’s systematic coverage across tasks, representation schemes, and model families, as well as its attention to both performance and latency across diverse datasets.
>
> This benchmark comprises **over 330 experimental configurations**—5 datasets × 44 models (Table 2), 5 datasets × 18 models (Table 3), and 5 datasets × 4 models (Table 7, supplementary materials)—spanning diverse datasets, model architectures, and experimental settings. The study encompasses the entire pipeline, from data preprocessing to training, evaluation, and analysis, and reflects over **nine months** of intensive research. Our primary objective is to provide a thorough comparison of the recommendation capabilities of LLMs and DLRMs, thereby offering insights to guide future research.
>
> We sincerely appreciate your recognition of the comprehensiveness and informativeness of our work. Below, we provide a point-by-point response to your concerns.
>
> ### W1. Other Metrics
>
> > Although the authors claim consistency, readers must probe supplementary materials to ensure robustness. Including at least one secondary metric in the main paper would facilitate quicker evaluation.
>
> We apologize for not making this information more accessible. Due to space constraints, we included the results in the supplementary material (Tables 5 and 6). In the final version, we will move representative secondary metrics (e.g., nDCG@10) into the main paper to enhance its comprehensiveness and completeness.
>
> ### W2. Cost-benefit Analysis
>
> > The efficiency section clearly shows that LLMs are far slower than DLRMs. However, there is no cost–benefit analysis, e.g., model‐size vs. incremental accuracy, that could guide practitioners in choosing a middle ground.
>
> Thank you for the insightful suggestion. To provide a cost–benefit analysis, we present selected results from Table 2 (MIND dataset) in the main paper. The table below compares performance (GAUC) and efficiency (GPU latency) across different model sizes.
>
> |Model (size)|GAUC|GPU Latency (ms)|
> |-|-|-|
> |(zero-shot) Qwen3-0.6B|0.4927|36|
> |(zero-shot) Qwen3-1.7B|0.5117|44|
> |(zero-shot) Qwen3-4B|0.5448|56|
> |(zero-shot) Qwen3-8B|0.6036|70|
> |(zero-shot) Llama3-8B|0.4904|65|
> |(fine-tune) Llama3-8B|0.7345|65|
> |DCNv2 (7M)      |0.6802|3.6|
> |DCNv2-emb (95M) |0.7167|5.3|
>
> The results indicate that while Llama3-8B achieves the highest accuracy, LLM-based methods are significantly slower in inference compared to DLRMs. Therefore, Llama3-8B is suitable only when maximum accuracy is required and efficiency is not a primary concern.
>
> Notably, DCNv2 (7M) ranks third despite having the smallest model size, and DCNv2-emb (95M), which leverages pretrained item content embeddings, delivers competitive performance with low latency. Overall, DCNv2-emb offers the best trade-off between performance and efficiency, making it a practical choice for industrial applications.

---

> > ### Comment · Reviewer_7Gns · 2025-08-02
> > **Further Concern**
> >
> > After carefully reviewing the other reviewers’ comments and the authors’ responses, I still have the following concerns:
> >
> > 1. The authors only discussed techniques such as model compression, knowledge distillation, and accelerated inference in an observational manner, without providing concrete, practical case studies or implementation examples. This lack of actionable demonstrations may undermine the perceived contribution and real-world applicability of their work.
> >
> > 2. Regarding fairness and privacy considerations, I would expect to see quantitative evaluations to substantiate their design choices and demonstrate the framework’s robustness, e.g., reporting AUC differences across user groups with varying activity levels, or an analysis of potential misuse risks after anonymizing user data.

---

> > ### Author Response · Authors · 2025-08-03
> > **Further Response (1/2)**
> >
> > Thank you for dedicating your time over the weekend to thoroughly review our rebuttal and for your swift response. We **sincerely appreciate the opportunity** to further address your additional concerns.
> >
> > ### 1. LLM Efficiency
> > > The authors only discussed techniques such as model compression, knowledge distillation, and accelerated inference in an observational manner, without providing concrete, practical case studies or implementation examples. This lack of actionable demonstrations may undermine the perceived contribution and real-world applicability of their work
> >
> > While improving LLM efficiency is not the primary focus of this work, we agree that including actionable demonstrations and case studies would be informative and add practical value. Accordingly, we present results below that demonstrate how **model quantization** can be employed to enhance LLM efficiency.
> >
> > PyTorch and HuggingFace Transformers natively support training and inference in 32- and 16-bit precision, but do not provide native support for 8- or 4-bit inference. Therefore, we utilize external packages such as "bitsandbytes" to enable 8- and 4-bit inference. The results presented in the tables below were obtained using a single A100 80GB GPU.
> >
> > **Zero-shot Performance**
> >
> > In the following experiment, we compare Qwen3-8B and its quantized variants across five datasets, evaluating both recommendation performance (GAUC) and efficiency (GPU memory usage and inference latency).
> >
> > |Model|Bit Precision|MIND&nbsp;&nbsp;|CDs&nbsp;&nbsp;|MicroLens|Goodreads|H&M&nbsp;&nbsp;&nbsp;&nbsp;|GPU Memory (MB) (batch size = 8)|Latency per Sample (second)|
> > |-|-|-|-|-|-|-|-|-|
> > |Qwen3-8B|32|0.5932|0.5172|0.6605|0.5490|0.6643|56,806|1.375|
> > |Qwen3-8B|16|0.6036|0.5166|0.6618|0.5471|0.6678|30,382|0.167|
> > |Qwen3-8B|8|0.5879|0.5243|0.6529|0.5474|0.6894|24,172|0.225|
> > |Qwen3-8B|4|0.5760|0.5239|0.6411|0.5424|0.6633|21,050|0.146|
> >
> > From the zero-shot results, we observe that quantized models (especially 16-bit) achieve comparable performance to their 32-bit counterparts, while offering **up to 10× faster inference**. Interestingly, on certain datasets (e.g., CDs), lower-bit models even outperform higher-bit models. One possible explanation is that, in the zero-shot setting, LLM parameters are **primarily optimized for general language tasks rather than the specific target recommendation task**. As a result, the 32- and 16-bit models retain high-precision details that may be irrelevant to recommendation, potentially introducing noise into the ranking process.
> >
> > We also conducted experiments comparing 32- and 16-bit precision on the H&M dataset. The results, reported using the GAUC metric, indicate that there are only minor differences in performance between the two precisions.
> >
> > ||Zero-shot|Fine-tune|
> > |-|-|-|
> > |BERT (float32)|0.5204|0.8701|
> > |BERT (bf16)|0.5210|0.8688|
> > |Llama3-8B (float32)|0.5444|0.8598|
> > |Llama3-8B (bf16)|0.5454|0.8606|
> >
> > Therefore, to accelerate both training and inference in our benchmark, **models larger than 7 billion parameters are fine-tuned using 16-bit precision**, in line with common practice (e.g., LC-Rec [1] and GenRec [2], based on their official GitHub implementation).
> >
> > **Fine-tuning Performance**
> >
> > In the following experiment, Qwen3-8B and Llama3-8B are fine-tuned using 16-bit precision. For inference, we evaluate the 16-bit models alongside their quantized 8-bit and 4-bit counterparts. All experiments are conducted on the Amazon CDs dataset.
> >
> > |Model|Bit Precision|GAUC&nbsp;&nbsp;&nbsp;&nbsp;&nbsp;|GPU Memory (MB) (batch size = 8)|Latency per Sample (second)|
> > |-|-|-|-|-|
> > |Llama3-8B|16|0.6344|25,766|0.101|
> > |Llama3-8B|8|0.6275|20,382|0.109|
> > |Llama3-8B|4|0.6176|16,322|0.097|
> > |DCNv2-emb (w/ Llama3 embedding)|32|0.5944|3164|0.00003|
> > |Qwen3-8B|16|0.5919|30,372|0.167|
> > |Qwen3-8B|8|0.5872|24,172|0.225|
> > |Qwen3-8B|4|0.5872|21,050|0.146|
> > |DCNv2|32|0.5577|762|0.00003|
> >
> > The results demonstrate that the quantized 8-bit and 4-bit models still outperform DLRMs in GAUC, highlighting the potential of LLMs for recommendation tasks. However, it is important to note that the 8-bit and 4-bit models do not yield the expected significant improvements in inference efficiency; in fact, the 8-bit model demonstrates slower inference than the 16-bit model. This can be attributed to the **overhead introduced by utilizing external, rather than native, quantization packages** for the 8-bit and 4-bit models.
> >
> > Nevertheless, our findings indicate that quantization can substantially reduce memory usage and, under appropriate implementation, improve inference efficiency in both zero-shot and fine-tuned scenarios. Furthermore, model quantization can be combined with other techniques—such as knowledge distillation, pruning, and KV-cache optimization—to further enhance efficiency. Although this rebuttal focuses solely on model quantization due to time and scope constraints, we believe it provides a concrete example that demonstrates the significant potential of this research direction.

---

> > ### Author Response · Authors · 2025-08-03
> > **Further Response (2/2)**
> >
> > ### 2. Fairness and Privacy
> >
> > > Regarding fairness and privacy considerations, I would expect to see quantitative evaluations to substantiate their design choices and demonstrate the framework’s robustness, e.g., reporting AUC differences across user groups with varying activity levels, or an analysis of potential misuse risks after anonymizing user data.
> >
> > **A. Fairness**
> >
> > Thank you for highlighting the importance of fairness and privacy considerations. We appreciate the opportunity to provide quantitative evaluations to further demonstrate the robustness of our framework.
> >
> > To address this, we conducted additional experiments to evaluate model performance across user groups with varying activity levels. In our experimental setting, **all test users are unseen during training**; therefore, we define user activity according to the **length of each user's historical sequence** in the test set. Based on this definition, we establish a threshold as follows:
> >
> > $$threshold = \frac{max\\_length + min\\_length}{2}$$
> >
> > Users with sequence lengths below the threshold are labeled as *inactive*, while those above the threshold are labeled as *active*. All experiments are conducted on the Amazon CDs dataset.
> >
> > We compute GAUC for each group separately and report the difference as Diff. = GAUC_active − GAUC_inactive.
> >
> > **Zero-shot Performance**
> >
> > | Model        | Inactive | Active | Overall|Diff.  |
> > |--------------|----------|--------|-----|---|
> > | OPT-1B       | 0.4996   | 0.5350 | 0.5011 | 0.0354 |
> > | Llama3-8B    | 0.5206   | 0.5232 | 0.5207 |0.0026 |
> > | Qwen3-4B     | 0.5011   | 0.5273 | 0.5022 | 0.0262 |
> > | Qwen3-8B     | 0.5153   | 0.5447 | 0.5163 | 0.0294 |
> >
> > **Fine-tuning Performance**
> >
> > | Model        | Inactive | Active | Overall | Diff.    |
> > |--------------|----------|--------|----|----|
> > | OPT-1B       | 0.5790   | 0.6002 | 0.5799 | 0.0212 |
> > | Llama3-8B    | 0.6184   | 0.6327 | 0.6190 | 0.0143 |
> > | Qwen3-4B     | 0.5842   | 0.6042 | 0.5850 | 0.0200 |
> > | Qwen3-8B     | 0.5994   | 0.6058 | 0.5997 | 0.0064 |
> > |DCNv2| 0.5677 |0.6071 | 0.5693|0.0394|
> > |DCNv2-emb (w/ Llama3 embedding)|0.5829|0.5886|0.5831|0.0057|
> >
> >
> > The results indicate that LLMs maintain consistent performance across both inactive and active user groups, particularly in fine-tuning scenarios, thereby supporting the fairness of our benchmarking framework. Notably, when DCNv2 is integrated with Llama3-generated item embeddings, the performance difference between the two groups decreases from 0.0394 to 0.0057. This finding suggests that the **rich semantic embeddings extracted by LLMs** (Llama3 in this case) can **effectively capture user preferences**, even for inactive users with sparse interaction histories.
> >
> > **B. Privacy**
> >
> > We thank the reviewer for bringing attention to this important issue. In recommender systems, it is a **well-established practice** to represent users solely based on their historical interaction sequences, without incorporating explicit user IDs. This methodology has been widely adopted in models such as SASRec [3], BERT4Rec [4], NRMS [5], and MINER [6]. Such an approach not only enhances user privacy but also **improves the generalizability and robustness** of the model, as it learns user preferences from behavioral patterns rather than relying on potentially sensitive or unique identifiers. Moreover, by omitting explicit user IDs, the risk of user re-identification is substantially reduced, thereby strengthening privacy protection in real-world applications.
> >
> > Additionally, our experimental setting ensures that **users in the test set do not appear in the training set**. As a result, incorporating user IDs would not contribute to improved model performance in our scenario.
> >
> > We will incorporate these analyses and the additional experiments into the revised version of our paper. Once again, we **thank you for your thoughtful engagement** in the review process. We would be happy to provide any further clarifications or additional results as needed.
> >
> > [1] Adapting Large Language Models by Integrating Collaborative Semantics for Recommendation
> >
> > [2] GenRec: Large language model for generative recommendation
> >
> > [3] Self-Attentive Sequential Recommendation
> >
> > [4] BERT4Rec: Sequential Recommendation with Bidirectional Encoder Representations from Transformer
> >
> > [5] Neural News Recommendation with Multi-Head Self-Attention
> >
> > [6] Multi-Interest Matching Network for News Recommendation

---

> > > ### Comment · Reviewer_7Gns · 2025-08-04
> > >
> > > Thank you for your further responses. I have no additional questions and recommend that this paper be accepted.

---

> > ### Author Response · Authors · 2025-08-04
> > **Thank You!**
> >
> > Dear Reviewer 7Gns,
> >
> > Thank you very much for your thoughtful review and positive recommendation. We greatly appreciate your time and valuable feedback throughout this process. We are **especially grateful for the constructive and engaging discussions**, which have made this a truly rewarding experience.
> >
> > With best wishes,
> > The Authors

---

### Official Review · Reviewer_noaL · 2025-07-02

**Rating:** 4
**Confidence:** 4

**Summary:**

This paper introduces RECBENCH, a benchmarking platform designed to comprehensively evaluate the performance of large language models (LLMs) as recommenders in comparison with traditional recommendation systems. Experiments conducted on datasets from five diverse domains involving up to 17 models reveal that LLM-based recommenders demonstrate significant performance gains in Click-Through Rate (CTR) prediction and sequential recommendation (SeqRec) tasks. However, their low inference efficiency limits their practicality for real-time recommendation scenarios. Meanwhile, the LLM-FOR-RS paradigm, which integrates LLM capabilities into conventional deep learning recommendation models (DLRMs), achieves a favorable trade-off between accuracy and inference speed, indicating strong potential for future development. Nevertheless, the study also highlights that LLMs may amplify data biases or reinforce user stereotypes, underscoring the importance of ethical auditing and cautious deployment in real-world applications.

**Dataset Code Accessibility:**

Yes

**Dataset Code Comments:**

The paper explicitly states that the code and datasets have been publicly released.

**Ethical Considerations:**

No, there are no or only very minor ethics concerns

**Final Justification:**

The authors addressed my questions, so I will keep my positive score.

**Limitations Weaknesses:**

1.	RECBENCH lacks considerations of fairness and privacy, which are critical in today’s data-driven recommendation research. In real-world applications, recommenders may produce biased outcomes for different user groups or raise privacy concerns. For instance, training on biased data can lead to unequal exposure or degraded recommendation quality for certain users, potentially reinforcing societal inequities.
2.	Despite its strengths in model and dataset diversity, RECBENCH’s experimental setup remains limited. It primarily focuses on CTR prediction and sequential recommendation, while overlooking other important tasks such as context-aware or multimodal recommendation, thus limiting the generalizability of its findings.

**Strengths Contributions:**

1.	RECBENCH offers a comprehensive evaluation protocol, covering diverse item representations, i.e., IDs, text, semantic embeddings, and semantic tags, to systematically assess their impact on recommendation performance. It supports both CTR prediction and SeqRec, making the evaluation more representative across various recommendation scenarios.
2.	The benchmark features a diverse set of models and datasets, including 17 LLMs ranging from general-purpose models (e.g., LLaMA) to recommendation-specific models (e.g., RecGPT), and spans five domains: fashion, news, video, books, and music. This diversity ensures broader applicability and more robust comparisons across tasks and domains.
3.	RECBENCH emphasizes both accuracy and inference efficiency, and supports both zero-shot and fine-tuning evaluation modes, enabling a thorough assessment of LLMs’ recommendation capabilities and adaptability.
4.	The platform aligns with the growing trend of applying LLMs in recommendation systems, offering not only a unified evaluation framework but also guiding future research directions， such as accelerating LLM inference and deeply integrating LLMs with traditional recommenders.

---

> ### Author Rebuttal · Authors · 2025-07-31
>
> Thank you for your positive feedback and recognition of RecBench’s comprehensive coverage of item representations, model diversity, and its dual focus on accuracy and efficiency.
>
> This benchmark comprises **over 330 experimental configurations**—5 datasets × 44 models (Table 2), 5 datasets × 18 models (Table 3), and 5 datasets × 4 models (Table 7, supplementary materials)—spanning diverse datasets, model architectures, and experimental settings. The study encompasses the entire pipeline, from data preprocessing to training, evaluation, and analysis, and reflects over **nine months** of intensive research. Our primary objective is to provide a thorough comparison of the recommendation capabilities of LLMs and DLRMs, thereby offering insights to guide future research.
>
> We sincerely appreciate your recognition of its value for informing future directions, such as LLM acceleration and hybrid recommendation systems. Below, we provide a point-by-point response to your concerns.
>
> ### W1. Fairness and Privacy
>
> > RECBENCH lacks considerations of fairness and privacy, which are critical in today’s data-driven recommendation research. In real-world applications, recommenders may produce biased outcomes for different user groups or raise privacy concerns. For instance, training on biased data can lead to unequal exposure or degraded recommendation quality for certain users, potentially reinforcing societal inequities.
>
> We thank the reviewer for highlighting these important issues. Fairness and privacy are indeed critical considerations in real-world recommender systems, and we have proactively incorporated these principles into the design of RecBench:
>
> - **User Split Strategy for Fairness**: To prevent unfair exposure bias toward highly active users, we ensure that each user appears exclusively in either the training or test set, but not both. This approach guarantees that all test users are unseen during training, reducing the risk of overfitting to popular users and promoting fairness in user representation.
>
>
> - **Privacy-Preserving Design**: To protect user privacy, we do not use sensitive attributes such as user ID, location, or age during training. Instead, models are trained solely on behavioral history sequences, requiring them to learn and generalize user interest patterns without relying on private information. This approach not only safeguards privacy but also enhances model robustness.
>
> ### W2. Other Recommendation Tasks
>
> > Despite its strengths in model and dataset diversity, RECBENCH’s experimental setup remains limited. It primarily focuses on CTR prediction and sequential recommendation, while overlooking other important tasks such as context-aware or multimodal recommendation, thus limiting the generalizability of its findings.
>
>
> Thank you for your thoughtful suggestion.
>
> We would like to emphasize that CTR prediction and sequential recommendation are the two main tasks in recommender systems, and our RecBench includes over **330** experimental settings covering both.
>
> RecBench currently focuses on textual content features, as these are the most widely used in the field. This focus is due to the difficulty of obtaining multimodal features for most datasets and the current limitations of LLMs in performing zero-shot recommendations with multimodal content.  However, our framework is flexible and can **support fine-tuning LLMs on multimodal recommendation data**. If multimodal features become available, RecBench **can be readily extended** to include experiments with them.
>
> Moreover, we use **users’ historical behavior sequences** -- rather than user IDs -- as the **primary context** for representing users. We also exclude other private user features, such as location and age. This design encourages the LLM to learn user interest patterns solely from behavioral data while also addressing the privacy concerns you raised in Weakness 1.

---

> > ### Author Response · Authors · 2025-08-05
> > **Additional Reference for Your Review**
> >
> > Dear Reviewer noaL,
> >
> > We hope you are having a pleasant week. We would like to kindly remind you that, when reviewing our rebuttal, you may also wish to refer to **our response to Reviewer 7Gns, titled “Further Response (2/2).”** In that section, we have provided a **more detailed analysis** that further addresses your concerns regarding **fairness and privacy**.
> >
> > Thank you very much for your time and consideration.
> >
> > Best regards,
> > The Authors

---

> > ### Author Response · Authors · 2025-08-07
> > **Thank You**
> >
> > Dear Reviewer noaL,
> >
> > Thank you very much for taking the time to read our rebuttal and for your kind acknowledgement. We are **grateful for your thoughtful feedback and valuable support**. If there are any further concerns or questions you would like to discuss, please do not hesitate to let us know.
> >
> > With best wishes,
> > The Authors

---

> > > ### Author Response · Authors · 2025-08-08
> > > **Kindly Share Your Final Evaluation**
> > >
> > > Dear Reviewer noaL,
> > >
> > > Please accept our apologies for reaching out to you again.
> > >
> > > We have just realized that, due to the current system, we are **unable to view the final justification you provided yesterday**, and you might be **unaware of this issue**. If possible, could you **kindly share your final evaluation of our work** and let us know if you have any remaining concerns? We would be more than happy to address any questions or provide any additional information you may need.
> > >
> > > Thank you very much for your time and consideration.
> > >
> > > With best regards,
> > > The Authors

---

### Official Review · Reviewer_PBJd · 2025-07-03

**Rating:** 5
**Confidence:** 2

**Summary:**

This submission presents RECBENCH, a comprehensive benchmark designed to systematically evaluate and compare large language model-based recommender systems (LLM-AS-RS) with traditional deep learning recommendation models (DLRMs). It explores multiple item representation forms—unique identifiers, text, semantic embeddings, and semantic identifiers—across two key recommendation tasks: click-through rate prediction and sequential recommendation. Evaluations cover a wide range of large models and diverse real-world datasets from various domains. The primary contributions include demonstrating the superior recommendation accuracy of large LLMs while highlighting their significant efficiency limitations, and providing an open, standardized platform to facilitate further research in improving both the effectiveness and efficiency of LLM-based recommender systems.

**Dataset Code Accessibility:**

Yes

**Ethical Considerations:**

No, there are no or only very minor ethics concerns

**Final Justification:**

I am still inclined to accept the paper.

**Limitations Weaknesses:**

1)  Although the paper highlights the significant inference inefficiency of LLM-based recommenders compared to traditional DLRMs (notably in Sections 4.2 and 5.2), the discussion remains mostly observational. The paper does not experimentally explore or propose concrete techniques for mitigating this bottleneck, such as model compression, distillation, or efficient inference strategies.

2) While RECBENCH covers five domains (fashion, news, video, books, music), it remains unclear whether these datasets sufficiently represent highly diverse user behaviors and item types, especially in industrial-scale or cross-domain recommendation scenarios (Section 3). Some domains, like social media or e-commerce, which are critical in practice, are not covered. Expanding the benchmark to include more varied datasets, especially large-scale industrial datasets or cross-domain recommendation tasks, would enhance its comprehensiveness and external validity.

3) The paper provides extensive quantitative evaluation (Tables 2 and 3), but lacks qualitative insights, such as examples of recommendations, failure cases, or interpretability analysis of LLM recommendations versus conventional models. Such analysis could offer deeper understanding of how LLMs leverage semantic information in recommendation. Including qualitative case studies or visualizations highlighting model behavior differences would enrich the paper’s contribution and make findings more tangible.

4) The semantic identifier approach, combined with conditional beam search, appears powerful but potentially complex and computationally intensive (Section 3.2). However, the paper does not provide detailed complexity analysis or scalability evaluation of this method. Adding a discussion or experiments focusing on the computational cost and scalability of semantic identifier decoding, compared to other methods, would help assess its practicality.

**Strengths Contributions:**

1) The introduction of RECBENCH as a unified evaluation platform is novel, particularly in its inclusion of diverse item representations—unique identifiers, text, semantic embeddings, and semantic identifiers—which provides a multifaceted perspective on how LLMs and conventional models handle recommendation tasks.

2) Its detailed investigation of the trade-off between the strong recommendation accuracy of large LLMs and their substantial efficiency challenges highlights an important practical concern, guiding future research priorities on inference acceleration (Section 4.2, 5.2, and 6).

3) By encompassing both zero-shot and fine-tuned settings, as well as different LLM architectures and scales, it aligns with current industrial trends and academic interests, especially given the growing use of LLMs in real-world recommender systems.

---

> ### Author Rebuttal · Authors · 2025-07-31
>
> Thank you for your positive feedback and for recognizing RecBench as a **unified evaluation platform** that incorporates diverse item representations. We also appreciate your acknowledgment of our detailed analysis of the accuracy-efficiency trade-off in LLMs and our comprehensive coverage of model settings and scales relevant to both academic and industrial contexts.
>
> This benchmark includes **over 330 experimental configurations**—5 datasets × 44 models (Table 2), 5 datasets × 18 models (Table 3), and 5 datasets × 4 models (Table 7, supplementary materials)—spanning a wide range of datasets, model architectures, and experimental settings. The study covers the full pipeline, from data preprocessing to training, evaluation, and analysis, and reflects over **nine months** of intensive research. Our primary objective is to provide a rigorous comparison of the recommendation capabilities of LLMs and DLRMs, offering insights to inform future research.
>
> We sincerely appreciate your recognition of RecBench’s value as a unified evaluation platform for advancing future research. Below, we provide a point-by-point response to your concerns.
>
> ### W1. Address LLM Efficiency Challenge
>
> > Although the paper highlights the significant inference inefficiency of LLM-based recommenders compared to traditional DLRMs (notably in Sections 4.2 and 5.2), the discussion remains mostly observational. The paper does not experimentally explore or propose concrete techniques for mitigating this bottleneck, such as model compression, distillation, or efficient inference strategies.
>
> We appreciate the reviewer’s comment and acknowledge this limitation of our work. Our primary focus was on comparing the performance and efficiency of LLMs and DLRMs; as such, we did not investigate techniques like model compression, distillation, or efficient inference strategies to address the efficiency challenges of LLMs. We agree that exploring these approaches would be a valuable extension to RecBench and plan to consider them in future work.
>
> ### W2. Dataset Coverage
>
> > While RECBENCH covers five domains (fashion, news, video, books, music), it remains unclear whether these datasets sufficiently represent highly diverse user behaviors and item types, especially in industrial-scale or cross-domain recommendation scenarios (Section 3). Some domains, like social media or e-commerce, which are critical in practice, are not covered. Expanding the benchmark to include more varied datasets, especially large-scale industrial datasets or cross-domain recommendation tasks, would enhance its comprehensiveness and external validity.
>
> We would like to clarify that the current five datasets in RecBench were deliberately selected to reflect a broad spectrum of real-world recommendation scenarios, including those mentioned by the reviewer:
> - MicroLens represents the video domain, specifically short video consumption in a **social media context**;
> - H&M represents the fashion domain, serving as a typical example of **e-commerce** recommendation;
> - CDs is sourced from Amazon and serves as an **industrial** dataset, capturing user-item interactions on a global online retail platform.
>
> To ensure consistency in the evaluation setup across datasets, we fix the training set size at 100,000, which is already a substantial scale. We agree that experiments with even larger-scale datasets would be valuable and plan to explore them in future work.
>
> We recognize that cross-domain recommendation is an important evaluation scenario. However, traditional DLRMs rely on ID-based item representations, making it difficult for them to generalize across datasets, as users and their IDs are typically not shared between domains. Therefore, DLRMs are not suitable for this task. Since our main goal is to compare DLRMs with LLMs, we did not include cross-domain evaluation in our main experiments.
>
> In contrast, LLM-based recommenders can transfer knowledge across domains by leveraging shared textual information. To explore this, we conducted preliminary experiments: we fine-tuned models on the MIND dataset and evaluated them on the test sets of other datasets, comparing these results to zero-shot performance (without fine-tuning). We observed positive cross-domain transfer on MicroLens, Goodreads, and CDs, but negative transfer on H&M. This intriguing effect warrants further investigation in future work.
>
> ||MIND|MicroLens|Goodreads|CDs|H&M|
> |-|-|-|-|-|-|
> |BERT-base (zero-shot)|0.4963|0.4992|0.4958|0.5059|**0.5204**|
> |BERT-base (fine-tune)|**0.7175**|**0.5877**|**0.5255**|**0.5128**|0.4932|
> |Llama-3 (zero-shot)|0.4904|0.5577|0.5191|0.5136|**0.5454**|
> |Llama-3 (fine-tune)|**0.7345**|**0.6104**|**0.5279**|**0.5176**|0.5235|
>
> ### W3. Qualitative Case Studies
>
> > The paper provides extensive quantitative evaluation (Tables 2 and 3), but lacks qualitative insights, such as examples of recommendations, failure cases, or interpretability analysis of LLM recommendations versus conventional models. Such analysis could offer deeper understanding of how LLMs leverage semantic information in recommendation. Including qualitative case studies or visualizations highlighting model behavior differences would enrich the paper’s contribution and make findings more tangible.
>
> Thank you for the insightful comment! Below, we present two case studies on the MIND dataset (for news recommendation) to illustrate how LLMs leverage semantic information in recommendation tasks. Each news article is represented by an ID beginning with “N” (e.g., “N50059”) along with its article title.
>
> - **Case One:**
>
> User click history:
> (1) N50059: *This Ford GT40 Movie Rig From "Ford V Ferrari" Looks Absurd*
> (2) N53017: *Kendall Jenner Wore the Tiniest Dress to Go Jewelry Shopping*
>
> The positive candidate item, N35729: *Porsche launches into second story of New Jersey building, killing 2*, receives a high relevance score from LLMs but not from DLRMs. This demonstrates that LLMs can capture deeper semantic connections—such as the shared theme of luxury cars (Ferrari in N50059 and Porsche in N35729)—which are often overlooked by traditional ID-based CTR models.
>
> - **Case Two:**
>
> User click history:
> (1) N35: *I Tried an Intense Metabolic Reset Program for a Month -- and It Worked*
> (2) N35452: *Potentially historic wind event' over weekend could inflame California wildfires*
>
> The positive candidate item, N42634: *The Latest Weight Loss Pills That Work And the Ones That Don't*, does not share any overlapping words with N35. Despite this, LLMs assign it a high relevance score by recognizing its relevance to the user’s interest in health and weight loss. This highlights the models’ ability to understand and generalize semantic relationships, rather than relying only on direct word matches.
>
> We will include these and additional case studies in the next revision of our paper.
>
> ### W4. Complexity Analysis on Conditional Beam Search (CBS)
>
> > The semantic identifier approach, combined with conditional beam search, appears powerful but potentially complex and computationally intensive (Section 3.2). However, the paper does not provide detailed complexity analysis or scalability evaluation of this method. Adding a discussion or experiments focusing on the computational cost and scalability of semantic identifier decoding, compared to other methods, would help assess its practicality.
>
> Based on the results in Table 3 of the main paper, we observe that CBS decoding is **only marginally slower** (approximately 3–5%) than standard beam search (BS) for large-size models. We provide the time complexity below:
>
> In standard BS, the model maintains a set of $B$ partial hypotheses (beams), expanding each with all $V$ vocabulary tokens at each decoding step. At step $t$, for each beam, the model outputs a logit vector of size $V$, from which the top-$k$ candidates are selected based on cumulative scores. This process is repeated for $T$ decoding steps. Therefore, the total time complexity of standard beam search is: $\mathcal{O}(B \cdot V \cdot T)$.
>
> Under the same setup, CBS maintains full vocabulary logits but applies a dynamic mask over invalid tokens $\mathcal{O}(B \cdot V \cdot T)$, which matches the complexity of vanilla BS in asymptotic terms. Ideally, a more efficient implementation could reduce the complexity to $\mathcal{O}(B \cdot \bar{F} \cdot T)$, where $\bar{F}$ is the average number of valid next tokens per Trie node (typically $\bar{F} \ll V$). But in practice, we retain full-vocabulary logits to **enable batched decoding**, and thus the time complexity remains $\mathcal{O}(B \cdot V \cdot T)$.
>
> Although CBS and standard beam search share the same theoretical time complexity, CBS is typically somewhat slower in practice. This slowdown is primarily due to additional operations such as dynamic logits masking and the overhead associated with Trie lookups. Nevertheless, given the performance gains relative to the modest increase in computational cost, the use of CBS remains well justified.
>
> We will incorporate the time complexity analysis in the next revision of our paper.

---

> > ### Author Response · Authors · 2025-08-05
> > **Additional Reference for Your Review**
> >
> > Dear Reviewer PBJd,
> >
> > We hope you are having a pleasant week. We simply wish to kindly remind you that, when reviewing our rebuttal, you may also wish to refer to **our response to Reviewer 7Gns, titled “Further Response (1/2).”** In that section, we have provided **case studies** that further address your concerns regarding **LLM efficiency**.
> >
> > Thank you very much for your time and consideration.
> >
> > Best regards,
> > The Authors

---

> ### Comment · Reviewer_PBJd · 2025-08-07
>
> I appreciate the author's efforts and responses. Most of my concerns have been addressed, and I am still inclined to accept the paper.

---

> > ### Author Response · Authors · 2025-08-07
> > **Thank You**
> >
> > Dear Reviewer PBJd,
> >
> > Thank you very much for taking the time to review our rebuttal. We are **delighted to know that our responses have addressed most of your concerns**. We sincerely appreciate your thoughtful feedback, your positive assessment, and your valuable support.
> >
> > With kind regards,
> > The Authors

---

### Official Review · Reviewer_duWa · 2025-07-06

**Rating:** 4
**Confidence:** 4

**Summary:**

This paper presents a comprehensive evaluation of four different item representation strategies, especially item IDs, raw text, semantic embeddings, and semantic identifiers, and their impact on recommendation performance in next-item prediction and item scoring tasks. The study examines both traditional deep learning-based recommender systems (DLRSs) and large language model-based recommender systems (LLM-RSs). Experimental results across five datasets demonstrate that fine-tuned LLM-based recommenders generally outperform traditional DLRSs. The authors have made both the code and data publicly available for reproducibility.

**Dataset Code Accessibility:**

Yes

**Dataset Code Comments:**

There are github repo and kaggle link provided for code and dataset included in the paper.

**Ethical Considerations:**

No, there are no or only very minor ethics concerns

**Final Justification:**

The authors have addressed all of my concerns, so I’m happy to update my rating to 4.

**Limitations Weaknesses:**

1. The paper lacks key experimental details, especially regarding the prompt tepleates on zero-shot prompting approach, fine-tuning setup, e.g. parameter configurations, and statistical significance testing. Since LLMs are highly sensitive to prompts, the absence of such details makes it difficult to fully assess the validity of the results. It is also recommended that the authors include few-shot experimental results for comparison.
2. The experiments are limited in scope, using relatively few state-of-the-art LLMs. Broader evaluation across diverse architectures and model sizes would strengthen the study’s conclusions.
3. The analysis focuses primarily on performance differences across models and representation strategies but pays insufficient attention to variations between datasets. A deeper exploration of how dataset characteristics influence outcomes—such as differences in content type, domain, or sparsity—is recommended.

**Strengths Contributions:**

The conclusions drawn regarding the influence of item representations across different models are informative and valuable for future research. Given the unique demands of recommender systems, particularly in time-sensitive domains such as news and short videos, the consideration of latency adds practical relevance to the findings.

---

> ### Author Rebuttal · Authors · 2025-07-31
>
> Thank you for your comments and for recognizing the value of our benchmark on item representation across various models and its contribution to future research directions.
>
> We would like to emphasize that our benchmark includes **over 330 experimental configurations**, encompassing 5 datasets × 44 models (Table 2), 5 datasets × 18 models (Table 3), and 5 datasets × 4 models (Table 7, supplementary materials). These configurations cover a wide range of datasets and models. Our study addresses the entire pipeline—from data preprocessing to model training, evaluation, and analysis—and reflects **over nine months** of intensive research. The primary aim is to provide a thorough comparison of the recommendation capabilities of LLMs and DLRMs, thereby offering valuable insights to guide future research in this field.
>
> We appreciate the opportunity to address your concerns, and hope our response will prompt a reassessment of our work and lead to a more favorable evaluation.
>
> ### W1: Missing Experimental Details
>
> Please note that the **implementation details and significance tests are provided in the supplementary materials**. We apologize for any inconvenience caused by their limited accessibility due to the space constraints.
>
> - **Finetuning Configurations:** they are provided in **Section C.1** of the supplementary material.
> - **Significant Testing:** they are also provided in **Section C.1**. All key comparisons report p-values less than 0.05.
> - **Prompting details:** A simplified prompt is illustrated in **Figure 1**.
>
> To further address your concern regarding  the prompt templates, we conducted experiments using the following prompts:
>
> |Version|Prompt|
> |-|-|
> |P1 (zero-shot)|You are a recommender. Please response "YES" or "NO" to represent whether this user is interested in this item. User history sequence: <user_history_sequence>. Candidate item: <candidate_item>. Answer (Yes/No):|
> |P2 (zero-shot)|You are a recommender. I will provide user behavior sequence, and a candidate item. Please response ‘YES’ or ‘NO’ to represent whether this user is interested in this item. You are not allowed to response any other words for any explanation or note. Now, your role formally begins. Any other information should not disturb you. User history sequence: <user_history_sequence>. Candidate item: <candidate_item>. Answer (Yes/No):|
> |P2 (1-shot)|You are a recommender. I will provide user behavior sequence, and a candidate item. Please response "YES" or "NO" to represent whether this user is interested in this item. \-\-\- Here is an example. User behavior sequence: (1) This Ford GT40 Movie Rig From "Ford V Ferrari" Looks Absurd, (2) Kendall Jenner Wore the Tiniest Dress to Go Jewelry Shopping. Candidate item: Donald Trump Jr walks out of Triggered book launch after heckling from supporters. Answer (Yes/No): Yes. \-\-\- Now, your role formally begins. Any other information should not disturb you. User history sequence: <user_history_sequence>. Candidate item: <candidate_item>. Answer (Yes/No):|
> |P2 (2-shot)|*Omitted due to space constraints*|
> |P2 (5-shot)|*Omitted due to space constraints*|
>
> Note: P2 (2-Shot) and P2 (5-Shot) follow the same format as P2 (1-Shot), but with more examples. Due to space constraints, they are not shown here.
>
> P1 is a concise prompt; P2 is more detailed; and P2 (1-shot) follows the in-context learning paradigm by including a demonstration example. The average GAUC scores across five datasets (MIND, MicroLens, Goodreads, CDs, and H&M) using the Qwen3-8B model under these prompts are summarized below:
>
> |Setting|MIND|MicroLens|Goodreads|CDs| H&M|
> |-|-|-|-|-|-|
> |**P1 (zero-shot)**|0.5847|0.6543|**0.5439**|0.5180|0.6499|
> |**P2 (zero-shot)**|0.6036|**0.6618**|0.5371|0.5175|**0.6678**|
> |**P2 (1-shot)**|0.6097 ± 0.0048|0.6254 ± 0.0043|0.5356 ± 0.0056|0.5056 ± 0.0038|0.6178 ± 0.0016|
> |**P2 (2-shot)**|**0.6354 ± 0.0050**|0.6536 ± 0.0056|0.5360 ± 0.0060|0.5150 ± 0.0069|0.6105 ± 0.0146|
> |**P2 (5-shot)**|0.6208 ± 0.0052|0.6565 ± 0.0071|0.5429 ± 0.0028|**0.5188 ± 0.0049**|0.6176 ± 0.0124|
>
> Overall, for zero-shot performance,  P2 offers only minor improvements over P1. Therefore, in our experiments, we used P1 for smaller models (size < 7B) with shorter input windows to conserve input tokens and allow for longer user sequences. For larger models with longer input windows, we used P2. Additionally, the use of fixed prompts aligns with previous LLM-based recommendation studies (e.g., TallRec [1], LC-Rec [2], and Uni-CTR [3]).
>
> The few-shot experiments were conducted three times, with each run using different examples randomly selected for in-context demonstrations. Compared to zero-shot performance, we observed that few-shot prompting does not always lead to improved results and can, in some cases, cause a significant drop in performance, as seen on the H&M dataset. This intriguing effect warrants further investigation in future work.
>
> We will incorporate these findings in the revised revision.
>
> [1] TALLRec: An Effective and Efficient Tuning Framework to Align Large Language Model with Recommendation
> [2] Adapting Large Language Models by Integrating Collaborative Semantics for Recommendation
> [3] A Unified Framework for Multi-Domain CTR Prediction via Large Language Models
>
>
> ### W2. Include Evaluation of More Recent Models
>
> We would like to clarify that our study already includes a broad selection of LLMs, ranging from early models such as BERT to recent state-of-the-art models released in 2025 (e.g., DeepSeek-V2-Qwen, released in **March 2025**). The model architectures covered in our benchmark include **OPT**, **LLaMA**, the **Qwen** series, and recommendation-specific LLMs like **RecGPT**. As shown in Table 1 of the main paper, our benchmark evaluates **17 foundation models**, making it the most comprehensive among existing RS+LLM benchmarks.
>
> The **Qwen3 series** was released in **late April** 2025, just two weeks before the NeurIPS submission deadline (May 15). Due to these time constraints, we were unable to include Qwen3 models in the main paper.
>
> Below, we provide zero-shot and fine-tuning results (fine-tuned on CDs) using the Qwen3 series models:
>
> |Model|MIND (zero-shot)|MicroLens (zero-shot)|Goodreads (zero-shot)|H&M (zero-shot)|CDs (zero-shot)|CDs (fine-tuned)|
> |-|-|-|-|-|-|-|
> |0.6B|0.4927|0.5165|0.5140|0.5993|0.5055|0.5098|
> |1.7B|0.5117|0.5704|0.5027|0.6571|0.4972|0.5775|
> |4B|0.5448|0.6394|0.5054|0.6487|0.5101|0.5879|
> |8B|0.6036|0.6618|0.5371|0.6678|0.5175|0.5919|
>
> The results demonstrate a clear trend consistent with scaling laws: model performance improves as model size increases, and fine-tuning yields significant performance gains. Overall, Qwen3-8B achieves zero-shot performance comparable to that of Qwen2-7B reported in the main paper.
>
> Additionally, we intentionally **omit models larger than 8B parameters**. This decision is primarily motivated by the **real-time constraints** of recommendation systems, where low-latency inference is essential. In industrial settings, 8B already represents the **practical upper limit for deployment in production environments**, and model compression or distillation techniques are typically required to ensure efficiency. We will incorporate experiments on the Qwen3 series in the revised version.
>
> ### W3. Explore Impact of Dataset Characteristics
>
> Thank you for the suggestion. Our primary goal in this work is to comprehensively evaluate the recommendation capabilities of both LLMs and DLRMs. To achieve this, we deliberately selected five datasets from diverse domains—fashion, news, video, books, and music—covering a wide range of recommendation scenarios, including e-commerce, social media, and industrial applications.
>
> Currently, our RecBench focuses on textual content features, as these are the most widely used in the field. This focus is due to two main reasons: first, it is often difficult to obtain multimodal features for most datasets; second, current LLMs have limitations in performing zero-shot recommendations with multimodal content. However, our framework is flexible and can support fine-tuning LLMs on multimodal recommendation data. If multimodal features become available, RecBench can be easily extended to include experiments with them.
>
> In general, domains that provide richer textual item features—such as news recommendation, where text is the primary modality—tend to benefit more from LLM-based recommenders. However, it is challenging to isolate the effects of content type, domain, and data sparsity when analyzing model performance.
>
> To address this, we conducted an experiment where we fixed the content type and domain by using a single dataset (CDs) and varied the sparsity level through downsampling. The results of this experiment are shown below:
>
> |Name|Train|Test|#Item|#User|Sparsity|Llama3-8B|DCNv2|QWen3-8B|
> |-|-|-|-|-|-|-|-|-|
> |CDs-small|50,000|20,000|72,796|12,355|99.99444%|0.6071|0.5463|0.5707|
> |CDs|100,000|20,000|113,671|24,602|99.99642%|0.6268|0.5664|0.5919|
> |CDs-large|150,000|20,000|146,604|36,916|99.99723%|0.6300|0.5714|0.6001|
>
> We use a standard definition of sparsity, defined as:
>
> $$sparsity = 1 - density = 1 - \frac{\\#train\_samples}{\\#items \times \\#users}$$
>
> As the number of training samples increases, data sparsity (as defined above) also increases. Despite this, the performance of Llama3-8B, Qwen3-8B, and DCNv2 consistently improves. This is because each user is represented by a sequence of item interactions, and with more item occurrences (i.e., more training samples), the item embeddings are trained more thoroughly and become more robust. Furthermore, the LLM-based models consistently outperform the DLRM (DCNv2) across all sparsity levels, demonstrating their effectiveness.
>
> We will incorporate these findings in the next revision of our paper.

---

> > ### Author Response · Authors · 2025-08-08
> > **Follow-Up Regarding Our Rebuttal**
> >
> > Dear Reviewer duWa,
> >
> > We hope you are having a pleasant week.
> >
> > Please accept our apologies for reaching out to you directly; we **do not wish to rush you in any way**. We fully understand and sincerely appreciate the significant commitment and effort that reviewing papers entails.
> >
> > However, as the reviewer-author discussion period **will conclude tomorrow**, we kindly ask if you could take a moment to review our rebuttal and let us know if you have any further questions or concerns. We would be more than happy to provide any additional clarifications or information you may require.
> >
> > Thank you very much for your time and consideration.
> >
> > With best regards,
> > The Authors

---

> > ### Comment · Reviewer_duWa · 2025-08-08
> >
> > Thanks for your reply! You’ve addressed all of my concerns, so I’m happy to update my rating to 4.

---

> > > ### Author Response · Authors · 2025-08-08
> > > **Thank You**
> > >
> > > Dear Reviewer duWa,
> > >
> > > Thank you very much for your response and for updating your rating. We **greatly appreciate your confirmation that our responses have resolved all of your concerns**. Your open-mindedness and support of our work are sincerely appreciated.
> > >
> > > Thank you again for your valuable feedback and consideration.
> > >
> > > With best wishes,
> > > The Authors

---

### Decision · Program_Chairs · 2025-09-18

**Decision:**

Accept (poster)

**Comment:**

This paper presents a benchmark comparing LLMs and deep learning-based recommender models (DLRMs) for two recommendation tasks: click-through rate prediction (CTR) and sequential recommendation (SeqRec). They consider 17 large models across 5 datasets.

All of the reviewers lean towards accepting this paper (scores: 4, 5, 4, 4), though they were (4, 4, 2, 4) before the rebuttal phase. Reviewer 7Gns championed the paper.

This work covers a lot of ground (different different prediction tasks, model architectures, datasets, etc.) but there is little discussion about the choice of hyperparameters; we only know from the tables that the LLM approaches give better accuracy (AUC). The latency columns in Table 2 and Table 3 are very nice to see.